# AI-derived 3D cloud tomography from geostationary 2D satellite data

Sarah Brüning[1], Stefan Niebler[2], and Holger Tost[1]

[1]Institute for Physics of the Atmosphere, Johannes Gutenberg University Mainz, Johann-Joachim-Becher-Weg 21, Mainz, 55128, Rhineland-Palatine, Germany
[2]Institute of Computer Science, Johannes Gutenberg University Mainz, Staudingerweg 9, Mainz, 55128, Rhineland-Palatine, Germany

**Correspondence:** Sarah Brüning (sbruenin@uni-mainz.de)

**Abstract.** Satellite instruments provide high-temporal-resolution data on a global scale, but extracting 3D information from current instruments remains a challenge. Most observational data is 2D, offering either cloud top information or vertical profiles. We trained a neural network (Res-UNet) to merge high-resolution satellite images from MSG SEVIRI with 2D CloudSat radar reflectivities to generate 3D cloud structures. The Res-UNet extrapolates the 2D reflectivities across the full disk of MSG SEVIRI, enabling a reconstruction of the cloud intensity, height, and shape in 3D. The imbalance between cloudy and clear-sky CloudSat profiles results in an overestimation of cloud-free pixels. Our RMSE accounts for 2.99 dBZ. This corresponds to 6.6 % error on a reflectivity scale between -25 and 20 dBZ. While the model aligns well with CloudSat data, it simplifies multi-level and mesoscale clouds, in particular. Despite these limitations, the results can bridge data gaps and support research in climate science such as the analysis of deep convection over time and space.

## 1 Introduction

Clouds and their interdependent feedback mechanisms have been a source of uncertainty in Earth system models for decades. Their influence to atmospheric gases and general circulation patterns is evident (Rasp et al., 2018; Shepherd, 2014; Bony et al., 2015). In a world affected by climate change, we require an accurate representation of cloud dynamics today more than ever (Norris et al., 2016; Stevens and Bony, 2013; Vial et al., 2013).

In recent years, observational data from remote sensing instruments has been used to investigate cloud properties on multiple scales (Jeppesen et al., 2019). Nevertheless, techniques to detect three-dimensional (3D) cloud structures on a large-scale are not yet established (Bocquet et al., 2015). Observations from passive sensors on geostationary satellites have a high spatio-temporal coverage, but they are limited to monitor the uppermost atmospheric layer in two-dimensions (2D) (Noh et al., 2022). By using the satellite's specificity at different wavelengths (Thies and Bendix, 2011), and subjective labeling or fixed thresholds (Platnick et al., 2017), we can estimate cloud physical properties like the cloud optical thickness (Henken et al., 2011) or the effective radius (Chen et al., 2020). In contrast, active radar penetrates the cloud top and delivers information on the subjacent reflectivity distribution (Barker et al., 2011). The radar receives detailed information on the cloud column along a 2D cross section with a high ground resolution and constant sun illumination. Due to its sun-synchronous orbit, it observes the same

spot at the same local time. Compared to geostationary satellites, the active radar does not provide a continuous spatial and temporal coverage (Wang et al., 2023). Passive sensors can be used to deliver an approximation of the cloud vertical column, but their information density is reduced compared to active sensors (Noh et al., 2022). Combining data sources can fill current data gaps (Amato et al., 2020; Steiner et al., 1995). The combined use of different instruments has been investigated before. This research comprises the usage of statistical algorithms (Miller et al., 2014; Seiz and Davies, 2006; Noh et al., 2022), the integration of radiative transfer approaches (Forster et al., 2021; Zhang et al., 2012), or the derivation of the multi-angle geometry of neighboring clouds (Barker et al., 2011; Ham et al., 2015) to reconstruct the cloud vertical column.

Emerging facilitators of data availability, like open-data policies and improved technological standards, enable effective processing of memory-consuming data (Irrgang et al., 2021; Rasp et al., 2018). This development promotes a further integration of computer science methods in climate science (Jeppesen et al., 2019; Liu et al., 2016). Ever-growing quantities of data surpass the capability of the human mind to extract explainable information efficiently (Lee et al., 2021; Karpatne et al., 2019). Here, the usage of artificial intelligence (AI) has been assigned a primary role (Runge et al., 2019). Cloud properties have been analyzed before by Machine Learning (ML) algorithms (Reichstein et al., 2019; Marais et al., 2020). The recent technological advances enable unprecedented operations (Amato et al., 2020). Deep-Learning (DL) based networks are suitable to identify spatial, spectral, and temporal patterns on big data (Jeppesen et al., 2019; Hilburn et al., 2020). In contrast to traditional ML frameworks, they do not require manual feature engineering (Le Goff et al., 2017). Adapting DL frameworks to applications in climate science offer new perspectives for a gain in knowledge (Rolnick et al., 2022; Jones, 2017).

So far, cloud properties have been investigated by DL algorithms in various applications. These comprise the detection (Drönner et al., 2018) and segmentation of cloud fields (Jeppesen et al., 2019; Lee et al., 2021; Le Goff et al., 2017; Tarrio et al., 2020; Cintineo et al., 2020), or the classification of distinct cloud types from meteorological satellites and aerial imagery (Marais et al., 2020; Wang et al., 2023). Zantedeschi et al. (2022) used a neural network to bring together information from an active radar and high resoluted satellite images to reconstruct cloud labels. Regressive models were used to predict rain rates (Han et al., 2022) or convective onset (Pan et al., 2021) for an improved weather forecast. These studies are often limited to reflect horizontal processes within the cloud field. Current studies by Hilburn et al. (2020) and Leinonen et al. (2019) use AI techniques such as convolutional neural networks (CNN) and conditional generative adversarial networks (CGAN) to address this issue. They reconstruct the 1D cloud column (Hilburn et al., 2020) or the 2D cross section of the input data (Wang et al., 2023). To our best knowledge, no extrapolation of 2D radar data to a large-scale 3D perspective was conducted before (Wang et al., 2023; Dubovik et al., 2021). Clouds move within a 3D space. This limits the prediction of multi-layer and mesoscale events by a 1D or 2D pixel-wise reconstruction (Hilburn et al., 2020). Models that do not consider the spatial coherence between pixels fail to reconstruct comprehensive cloud structures (Hu et al., 2021). Image segmentation approaches like the UNet (Ronneberger et al., 2015; Jiao et al., 2020; Wieland et al., 2019) may reconstruct the ground truth data more adequately. They can be used to provide the indicators for predicting clouds in 3D with its adjacent boundaries, shadow locations, and geometries (Wang et al., 2023). This can lead to a more realistic representation of the predicted clouds (Jiao et al., 2020).

In this study, we employ a modified Res-UNet (Diakogiannis et al., 2020; Hu et al., 2021) to integrate 2D data from active (polar orbiting satellite, radar) and passive (geostationary satellite, spectrometer) instruments to reconstruct a 3D cloud field.

Previous studies focused on reconstructing the 1D cloud column or 2D cross section. In contrast, our approach utilizes a DL framework to predict the radar reflectivity not only along the radar cross section but across the entire satellite full disk (FD). We use the radar height levels to extend 2D satellite channels into a 3D perspective. The goal is to establish a spatio-temporal consistent cloud tomography solely based on observational data. Predicted reflectivities can enhance the availability of 3D resolved cloud structures, particularly in regions with limited data.

## 2 Methods

### 2.1 Data overview

Our approach uses observational data from two different remote sensing sensors to predict a 3D cloud tomography. The input data for the neural network originates from a geostationary satellite. We use data from the European Organisation for the Exploitation of Meteorological Satellites (EUMETSAT) Spinning Enhanced Visible and InfraRed Imager (SEVIRI) instrument on the Meteosat Second Generation (MSG) satellite (EUMETSAT Data Services, 2023). This sensor observes the Earth from a height of 36000 km and provides 2D satellite images in a high spatial and temporal resolution. The ground truth of the study is derived from an active radar on board the CloudSat satellite which moves on a sun-synchronous orbit (CloudSat Data Processing Center, 2023). The 2D profiles along the track contain information on the cloud reflectivity. In our study, we feed the MSG SEVIRI data into a neural network to reconstruct the CloudSat radar reflectivity and extrapolate the 2D profiles to a 3D perspective.

### 2.1.1 Geostationary satellite images

Satellite images from the MSG SEVIRI instrument display the input for the network (later referred to as "imager data") (Schmetz et al., 2002). Observing the Earth's surface in intervals of 15 min and with a spatial resolution of 3 km at nadir, MSG SEVIRI provides information in 12 channels centered within wavelengths from 0.6–132 $\mu$m (Benas et al., 2017). Depending on the wavelength and daytime of retrieval, the channels are sensitive to reflected solar radiation or surface emissions (Table 1). They can be applied to approximate cloud physical properties (Sieglaff et al., 2013). Our approach uses 11 channels. The HRV channel is excluded due to its different resolution and uncertain added value. Three of the channels are sensitive to solar radiation, which restricts us to using only daytime data. We reformat all imager data onto a spatial grid with geographic coordinates, employing the global reference system WGS84 (Drönner et al., 2018). Each pixel has a resolution of 0.03 ° in both width (W) and height (H). To account for diminishing accuracy from the equator to the poles, we exclude the areas near the sensor boundaries (Bedka et al., 2010). The designated area of interest (AOI) extends 60 ° in all directions, marking the boundaries of the new FD.

**Table 1.** Overview of MSG SEVIRI channels (Schmetz et al., 2002).

| Channel | Center ($\mu$m) | Range ($\mu$m) | Type |
|---------|-----------------|----------------|------|
| VIS0.6 | 0.635 | 0.56-0.71 | Solar reflective |
| VIS0.8 | 0.81 | 0.74-0.88 | Solar reflective |
| NIR1.6 | 1.6 | 1.5-1.78 | Solar reflective |
| IR3.9 | 3.92 | 3.48-4.36 | Both |
| WV6.2 | 6.25 | 5.35-7.15 | Thermal IR |
| WV7.3 | 7.35 | 6.85-7.85 | Thermal IR |
| IR8.7 | 8.70 | 8.30-9.10 | Thermal IR |
| IR9.7 | 9.66 | 9.38-9.94 | Thermal IR |
| IR10.8 | 10.8 | 9.80-11.80 | Thermal IR |
| IR12.0 | 12.0 | 11.00-13.00 | Thermal IR |
| IR 13.4 | 13.4 | 12.40-14.40 | Thermal IR |
| HRV | n/a | 0.5-0.9 | Solar reflective |

### 2.1.2 Radar data

Within the CloudSat (CS) GEOPROF-2B product, a nadir-looking 94 GHz cloud profiling radar (CPR) delivers information on the cloud reflectivity in logarithmic dBZ scale (later referred to as "radar data") (Stephens et al., 2008). The radar receives a 2D cross section of the cloud column with a horizontal resolution of 1.1 km. The vertical dimension (Z) comprises 125 height levels with a bin size of 240 m (Guillaume et al., 2018). From the ground surface to the lower stratosphere, the vertical extent covers 30 km. We use the reflectivity transects as the ground truth to train and evaluate the model. In the subsequent steps, we adjust the height levels of the radar. The lower altitudes, specifically those between 0 and 3 km, are influenced by the topography and a radar signal weakening due to attenuation (Marchand et al., 2008). To enhance the model performance, we omit the lowest 10 height levels. Since we notice a significant imbalance between clear-sky and cloudy pixels, we exclude the predominantly cloud-free areas within the upper 25 height levels (Stephens et al., 2008). The final Z-dimension encompasses 90 height levels ranging from 2.4 km to 24 km. We note that due to the sun-synchronous orbit of CloudSat, it has a reduced ability to account for diurnal variations within specific regions of the AOI (Stephens et al., 2008).

### 2.1.3 Matching scheme

We obtain training data for our study by aligning MSG SEVIRI scenes with CloudSat radar data as shown in Figure 1 (a). To match the datasets, we compare their timestamps and locations. If the radar coordinates fall within the AOI, we determine the flight direction to identify whether CloudSat circles the Earth in ascending or descending orbit. We then extract images of 128

x 128 [H x W] pixels from each MSG SEVIRI channel along the radar coordinates using a moving-window approach with a 50 % overlap between image-profile pairs (Denby, 2020; Jeppesen et al., 2019).

We prepare the matched image-profile pairs for further processing. To do this, we combine the 11 MSG SEVIRI channels into a single 3D array with dimensions 11 x 128 x 128 [C x H x W] pixels. CloudSat flies across a horizontal transect within the satellite scene. It has a higher native resolution than MSG SEVIRI. To align the datasets, we downsample the radar pixels by aggregating them based on the local maximum reflectivity. This adjusts the CloudSat pixels to the MSG SEVIRI resolution of 0.03 ° but leads to some loss of sharp contrast in radar pixels (Jordahl et al., 2020). We standardize the data shape by

transforming the 2D cross section into a sparse 3D array of 125 x 128 x 128 [Z x H x W] pixels, representing reflectivities along the cross section. After downsampling, the transect becomes one-pixel wide. We label pixels outside the transect as missing values to maintain the CloudSat data's location during training. We use these pixel indices to compute the loss between the CloudSat data and the predicted cross section and to evaluate the model performance.

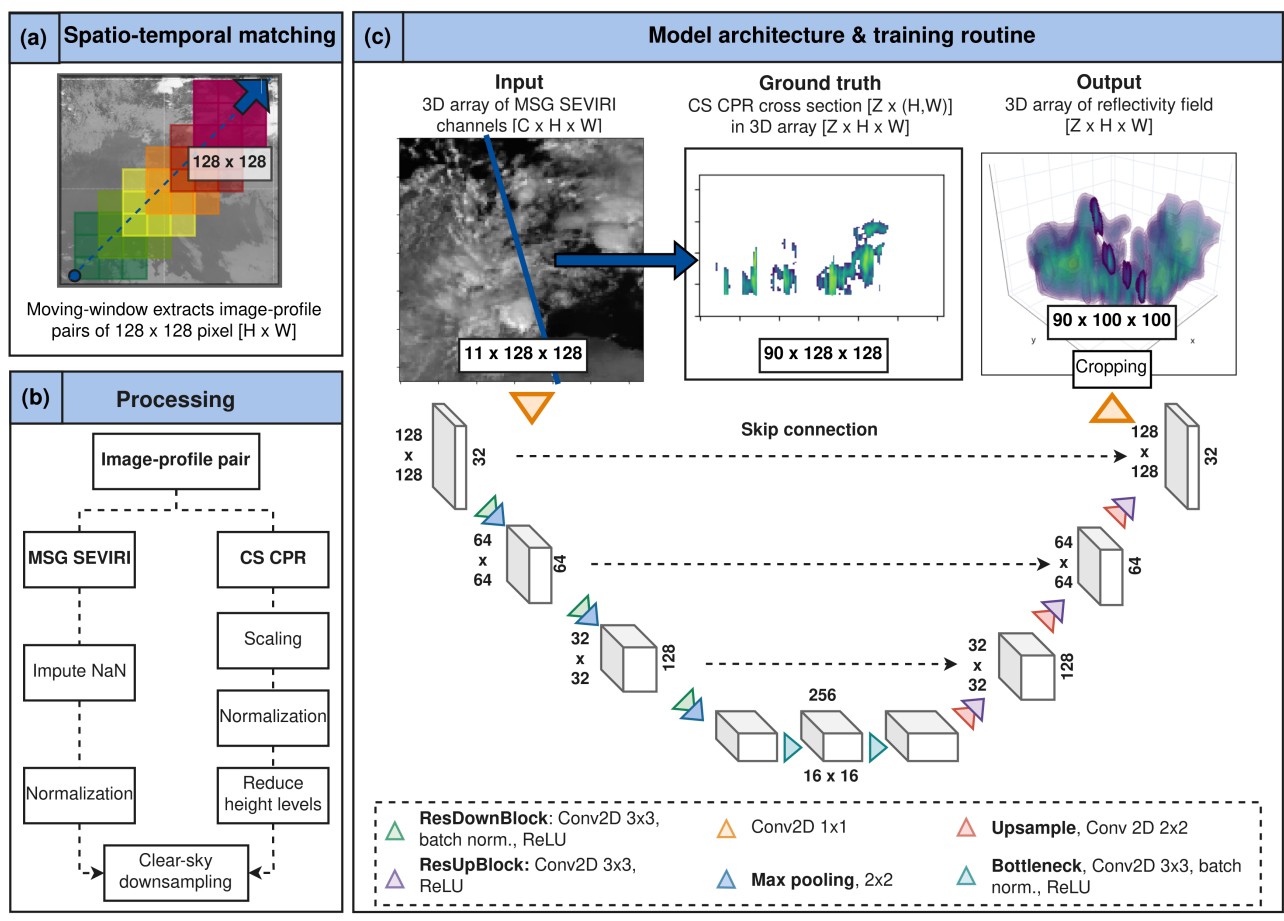

**Figure 1.** Workflow of the study. Part **(a)** points out the moving-window approach used for matching the radar and the imager data. Steps needed for the processing of both datasets are depicted in **(b)**. In **(c)**, the architecture of the proposed Res-UNet is illustrated. The upper row visualizes an example for the input data, ground truth (with reduced 90 height levels and full transparency for values <= -25 dBZ), and output respectively. The location of the radar transect within the 3D output image is pictured with full opacity. The numbers alongside the boxes in the architecture sketch refer to the feature channels (right) and image sizes (left) at the given model depth.

### 2.1.4 Data processing

Before training the model, we process the extracted image-profile pairs. We utilize a full year of data (2017) to incorporate seasonal variations into the modeling process. We split the 30000 matched image-profile pairs, with 75 % (January to September) used for training and 25 % (October to December) for validation. Our test set is derived from data in May 2016, from which the matching algorithm extracts 1500 image-profile pairs. We impute missing data in the 3D MSG SEVIRI array by an interpolation of neighbouring pixels (Troyanskaya et al., 2001). Afterwards, data from each satellite channel $x$ was normalized between [0,1] by

$$x' = \frac{x - \mu}{\sigma} \tag{1}$$

using the arithmetic mean $\mu$ and standard deviation $\sigma$ of the training data (Leinonen et al., 2019). As described in Section 2.1.2, we reduce the height levels of the CloudSat profile from 125 to 90 (Fig. 1, b). We use the CloudSat quality index to identify noisy pixels. Pixels with a quality index lower than six were set to a background value of -25 dBZ to reduce noise (Marchand et al., 2008). All radar reflectivity values $Z_{dB}$ were normalized between [-1,1] as follows

$$Z'_{dB} = 2\frac{Z_{dB} + 35dB}{55dB} - 1 \tag{2}$$

by the maximum and minimum reflectivity between [-35, 20] (Stephens et al., 2008; Leinonen et al., 2019). The CloudSat data is highly skewed towards clear-sky samples. We limit the percentage of cloud-free profiles to 10 % to tackle this imbalance (Jeppesen et al., 2019).

### 2.2 Model architecture and training

Neural networks can capture highly complex relationships between input and output data (Lee et al., 2021). The Res-UNet used in this study displays a modified framework designed for remote sensing data (Dixit et al., 2021). Additional residual connections and continuous pooling operations aim to reduce the dependence of the network on the input's location (Diakogiannis et al., 2020). Former studies using the Res-UNet dealt with the classification of tree species (Cao and Zhang, 2020) or the prediction of precipitation (Zhang et al., 2023). Obtained results emphasize the ability of the Res-UNet to adequately address the importance of the spatial coherence in environmental research (Marais et al., 2020). In this study, we derive the cloud reflectivities in dBZ from the satellite channels by a regression task (Hilburn et al., 2020; Zhang et al., 2023).

As introduced by Ronneberger et al. (2015), the UNet and its modifications provide an almost symmetrical architecture. The network architecture of the Res-UNet is shown in Figure 1 (c). The parameters of the network are listed in Table A1 in Appendix A. Each box represents the layer sizes on the encoder and decoder side. On the right side of each box, the filter size is given. The respective height and width are given on the left side.

The Res-UNet consists of six residual blocks, each including two 2D convolutions (3 x 3 kernel, stride 1) and ReLU activation (Diakogiannis et al., 2020). On the encoder side, we add batch normalization. The output is merged with a skip connection

that consists of one 2D convolution (3 x 3 kernel, stride 1), and a batch normalization. Adding the skip connection and the convolutional layer represents the output of a residual block.

We increase the channel dimension of the initial imager data from 11 x 128 x 128 pixels with a 1 x 1 2D convolution to a feature map of size 32 x 128 x 128. In the encoder branch, we then employ a sequence of three residual blocks with doubling filter sizes, each followed by a 2 x 2 max pooling layer (Lee et al., 2021). We subsequently reduce the feature map size to 256 x 16 x 16 pixels in the bottleneck layer. Here, we apply two 2D convolution layers followed by batch normalization and ReLU activation.

The decoder side features three residual blocks, each with an upsampling layer (2D convolution, 2 x 2 kernel, stride 2) and a corresponding skip connection from the encoder. After upsampling, we apply a residual block with 2D convolution (3 x 3 kernel, stride 1) and ReLU activation, doubling the spatial extent to match the skip connection while halving the channel dimension (Li et al., 2018). The final 1 x 1 convolution maps the output to 90 x 128 x 128 pixels, representing the 90 height levels of the radar cross-section (Jeppesen et al., 2019). We remove the border pixels of the output, resulting in a final radar reflectivity output of 90 x 100 x 100 pixels [C x W x H]. Predicted reflectivities are scaled between -35 and 20 dBZ, with values below -25 dBZ considered as cloud-free (Leinonen et al., 2019).

We conducted the training for 50 epochs with a batch size of 4 and a weight decay of 0.00001 (see Table A1 in Appendix A). We have 1893328 total trainable parameters. The estimated total size of the model is 194.27 MB (see Table B1 in Appendix B). We use the Adaptive Moment Estimation (ADAM) method for model optimization due to its fast convergence rate (Kingma and Ba, 2014). The learning rate is initially set to 0.001 (see Table A1 in Appendix A). It is reduced by a learning rate scheduler during the training process when reaching a plateau. To enhance the amount of training data, we give all input data a chance of 25 % to be rotated by 90 ° (Jeppesen et al., 2019). These flipped images are perceived as new samples. The goal is to increase the model invariance to the orientation of the radar cross section.

## 2.3 Evaluation

### 2.3.1 Analyzing and comparing the model performance

The model performance is quantified during training (loss function) and evaluated afterwards by calculating the root-mean-square error (RMSE) (see Table A1 in Appendix A). The RMSE is equally able to penalize misses and false alarms (Lee et al., 2021). As described in Section 2.1.3, we preserve the pixel indices of the CloudSat cross section within each image-profile pair during training. We use the location of these pixels to filter the observed and predicted transect. The RMSE is calculated along the filtered cross sections. Since it is only evaluated on a small subset of 10 % of all pixels, we have a sparse regression task (Wang et al., 2020). We cannot quantify the model performance on the full 3D prediction of the cloud field.

The results of the Res-UNet are compared against two competitive methods (Drönner et al., 2018). First, we predict the radar reflectivity by an ordinary least squares model with multiple regression output (OLS). The 11 satellite channels were used as independent predictor variables. The output is a 1D cloud column. Second, a Random-Forest (RF) regression is applied (Breiman, 2001). The RF is a supervised ML algorithm suitable when working with environmental datasets in the natural

sciences (Boulesteix et al., 2012). Its feasibility for complex meteorological data was investigated before, e.g., McCandless and Jiménez (2020) used a RF regression to detect clouds. Our study uses a setup with 100 trees, each choosing a random subset of satellite channels to predict the reflectivity along a 1D cloud column. We use the same training, validation and test split as for the Res-UNet. For each image-profile pair, we filter the 3D array to locate the radar cross section. This transect is separated into 1D cloud columns. For every pixel along the cross section, we receive a ground truth in the shape of 90 x 1 [Z x (H,W)]. The 3D array containing the satellite channels was filtered by the radar profile location and divided into images of the size 11 x 1 [C, (H,W)]. The OLS and RF map the imager data to an output size of 90 x 1 pixels [Z x (H,W)]. We calculate the RMSE between the observed and predicted cloud column and scale the output between -35–20 dBZ. We reconstruct the 2D transect by the preserved index of each pixel of the cross section. These profiles are compared to the output of the Res-UNet.

### 2.3.2 Merging 3D reflectivities on the FD

We predict the radar reflectivity for each MSG SEVIRI file in the test dataset (May 2016) using the trained Res-UNet. The result is a contiguous 3D cloud tomography for every 15 minute time step. The MSG SEVIRI FD covers an extent of 2400 x 2400 pixels. For the FD prediction, we divide the FD into overlapping subsets of 128 x 128 pixels. These subsets are processed and fed into the network. The output is a 3D reflectivity image of 90 x 100 x 100 pixels [Z x H x W], which equals 2.5 ° on the MSG SEVIRI grid. We merge the tiles to cover the whole satellite AOI. Between the tiles, there is no overlap. The goal is to evaluate the network's ability to extrapolate a large-scale cloud field from single tiles.

### 2.3.3 Computing the cloud top properties

To our best knowledge, there exist no comparable datasets to the 3D cloud tomography in this study. Instead of a quantitative evaluation of the reflectivity, we evaluate the predictions based on their performance to derive the cloud top height (CTH) (Wang et al., 2023). We use the FD predictions for the test dataset (May 2016) for the computation. The CTH is defined as the distance between the ground surface and the uppermost cloud layer for every 1D vertical column (Huo et al., 2020). This calculation requires converting the height levels into a kilometer scale. We use a fixed threshold of -15 dBZ to differentiate a cloudy from a clear-sky pixel (Marchand et al., 2008). The result is a binary classification for each pixel in the 3D cloud field. On this dataset, we extract the CTH as the top cloud signal on each 1d cloud column of the FD. We aggregate the results on a monthly scale and compare the predicted CTH to the operational product CLAAS-V002E1 (CLoud property dAtAset using SEVIRI, Edition 2) (Finkensieper et al., 2020). It is based on the MSG SEVIRI channels and additional model data and provides information on the macrophysical and microphysical cloud properties. We use a monthly aggregate with a resolution of 0.05 ° on the MSG SEVIRI FD.

 ## 3   Results

### 3.1   Evaluating the reflectivity distribution

We analyze the ability of the three models (Res-UNet, OLS, and RF) to reconstruct the cloud vertical distribution for the test dataset in May, 2016 (Sect. 2.3.1). The OLS and RF predict a 1D column whereas the output of the Res-UNet comprises a 3D image of the cloud field. We filter all outputs by the preserved location of the radar cross sections to derive the original 2D transect. At first, we compute the height dependent reflectivity distribution of the CloudSat data and the three models. Due to the applied quality flag (Sect. 2.1.4), we have few observations below 5 km height (Fig. 2, a). This leads to a shift in low height levels. The models overestimate cloud-free values below -25 dBZ. The CloudSat reflectivities have a peak at 0–10 dBZ between 5–7 km. A second, weaker peak is observed between 12–15 km for reflectivities < 0 dBZ. All predictions underestimate the first peak > 0 dBZ. Instead, they overestimate the occurrence of reflectivities < -20 dBZ (Fig. 2, b–d). The OLS shows an especially high shift towards low reflectivities. The Res-UNet predicts low reflectivities < -20 dBZ along all height levels between 5–15 km whereas we observe a distinct peak at 5 km for the RF.

We analyze the difference between the observed and predicted reflectivities by a two-dimensional joint distribution plot. For this purpose, we calculate the density distribution of the reflectivity between 2.4–24 km. Here, we use a bin size of 1 dbZ and 240 m height, respectively (Steiner et al., 1995). All distributions are calculated on the test dataset and normalized by the distribution size (n = 1500). Predictions differ from the original radar data, especially for values > 0 dBZ and in low altitudes (Fig. 3). In the joint plot, the highest agreement appears in the shape of a curved line between low reflectivities > 15 km to high reflectivities at 7 km. The results indicate an overestimation of high reflectivities and an underestimation of low reflectivities, especially for low-level clouds. Since we observe few clouds in high altitudes (Fig. 2, a), the distribution differences get smaller above 15 km. The joint plot shows a similar distribution for the Res-UNet and the RF, whereas the error of the Res-UNet is slightly lower for reflectivities between -15–0 dBZ (Fig. 3, a, c). We observe few predictions > 0 dBZ and a strong overestimation of reflectivities < -20 dBZ for the OLS (Fig. 3, b).

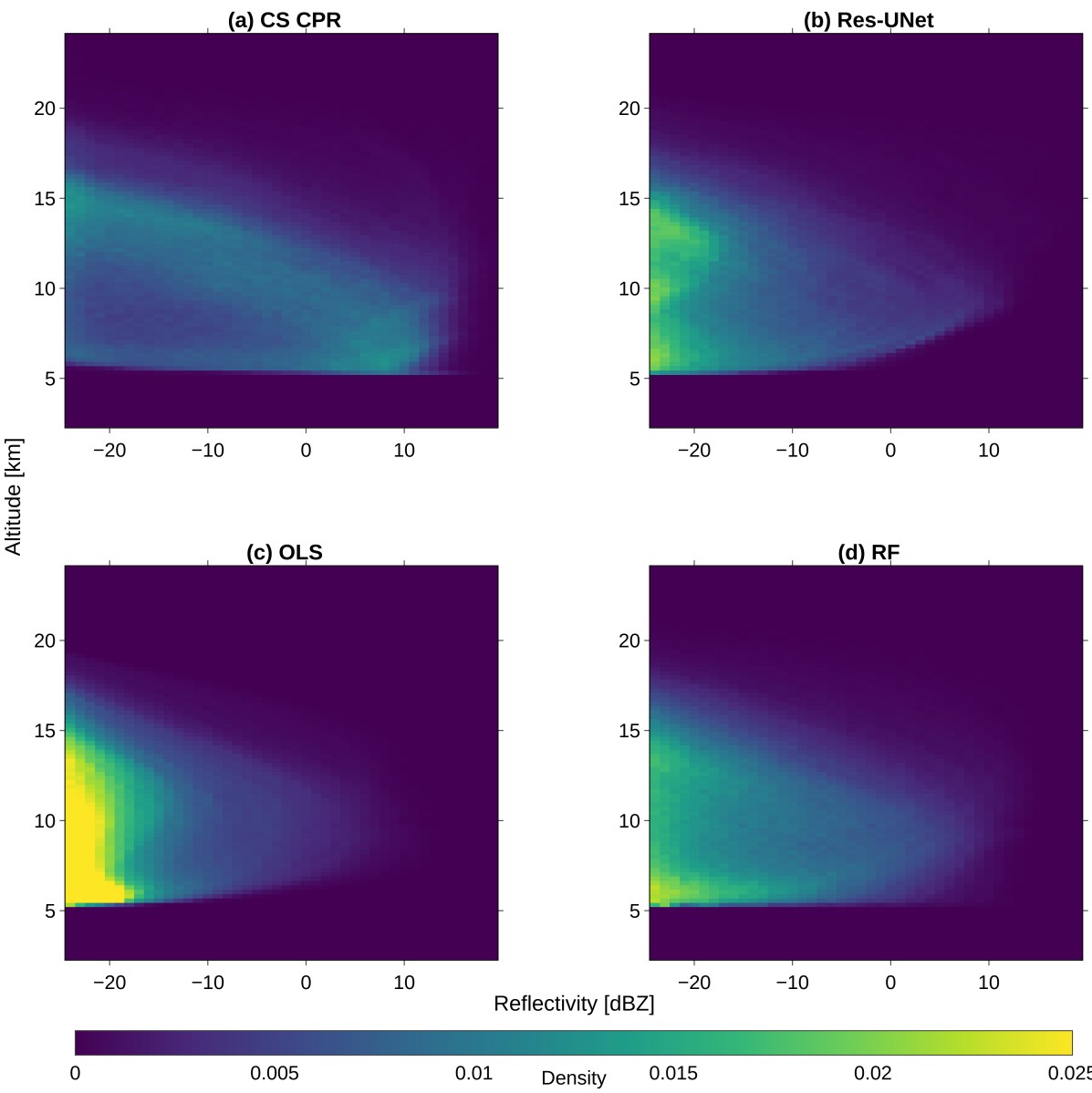

**Figure 2.** Height dependent reflectivity distribution for every height bin between 2.4–24 km for the CloudSat data (CS CPR) **(a)**, the Res-UNet **(b)**, the ordinary least squares model (OLS), **(c)**, and the Random-Forest regression (RF) **(d)** (n = 1500).

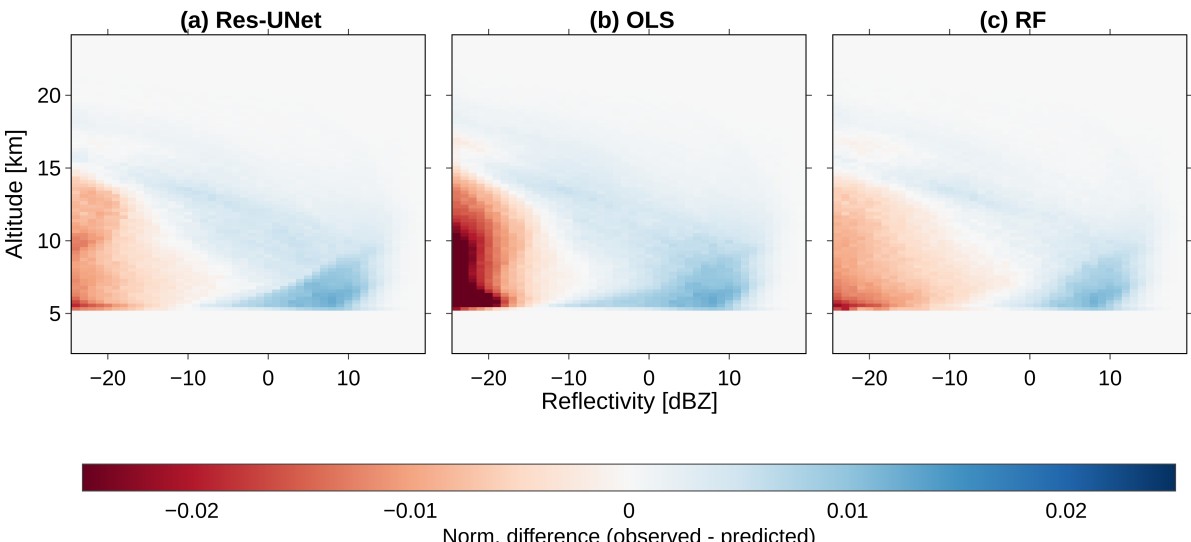

**Figure 3.** Joint plot of the normalized difference between the observed and predicted reflectivity on the test dataset (n = 1500). For each height bin between 2.4–24 km, we compare the CloudSat density distribution (CS CPR) to the distribution of the Res-UNet **(a)**, the ordinary least squares model (OLS) **(b)**, and the Random-Forest regression (RF) **(c)**.

## 3.2 Height-dependent model performance

We analyze the model error (RMSE) along the vertical cloud column. For all models, we calculate the mean RMSE on the test dataset between 2.4–24 km. The results point out an overall lower RMSE for the Res-UNet than for the OLS and RF (Fig. 4). The mean RMSE varies between 2.99 dBZ for the Res-UNet, 4.1 dBZ (RF), or 4.58 dBZ (OLS). On a dBZ scale between -25–20 dBZ, this is equivalent to an error of 10.1 % (OLS), 9.1 % (RF), or 6.6 % (Res-UNet). Between 2.4–5 km, the RMSE is 0. This is due to the lack of CloudSat observations after filtering noisy pixels (Fig. 2, a). Between 5–7 km, the RMSE increases to up to 8 dBZ for the Res-UNet, 10 dBZ for the RF, and 12 dBZ for the OLS (Fig. 4). In higher altitudes, the performance of all models improves. The RMSE decreases to 4 dBZ (5.7 dBZ, 6 dBZ) for the Res-UNet (RF, OLS) at 15 km and reaches its minimum at 22 km (24 km for OLS and RF). Above 15 km, we have few CloudSat observations > 15 dBZ (Fig. 2, a). We observe a lower model error (Fig. 4) and reduced difference between the distributions (Fig. 3) in these height levels for all three models. The improved performance can be led back to the superior number of background reflectivities or the presence of more uniform clouds, like extended tropical cirrus. Over all height levels, the Res-UNet has the lowest RMSE of the three models. Compared to the OLS (RF), the mean RMSE of the Res-UNet is reduced by 34,8 % (27,1 %).

Figure 5 shows the predicted and observed reflectivity along the radar transect for four randomly chosen samples. For all models, the reconstructed cloud signal is predicted at the right horizontal location along the cross section. Clear-sky situations of -25 dBZ are recognized without noise. The cross sections in Figure 5 (a) are created using processed CloudSat reflectivities with a resolution of 0.03 °. Although the radar pixels lose some sharp contrasts after the downsampling, we observe a higher blurriness for the predictions. The edges of individual clouds smear out for all three models. Even though all transects were labelled as cloudy, we see a high percentage of background pixels.

For the Res-UNet, we observe a RMSE between 3.3 and 8.2 dBZ. The overall shape and increased intensification towards the cloud's core follow the radar, even though edges are blurred, and peak reflectivities remain underestimated (Fig. 5, b). This issue is reflected within the reflectivity distribution of the DL model (Fig. 2, b). While the Res-UNet accurately identifies single-layer clouds, it misses sharp edges of multi-layer clouds, especially in mid-altitudes. Clouds over multiple height levels are blurry and show a reduced small-scale accuracy in the vertical dimension (Fig. 5, III). The lower height levels of multi-layer clouds are only partly represented (Fig. 5, II, IV). Instead, we observe a simplification of these cloud layers.

For the OLS and the RF, the underestimation of the cloud core reflectivities resembles the Res-UNet (Fig. 5, II, III). All four examples show a higher RMSE for the OLS and RF than for the Res-UNet (Fig. 5, c, d). The difference varies between 0.1 (I) and 2.7 (IV) dBZ. While the error is predominantly similar for all three models, the shape of the predicted clouds differs (I, III). The OLS (RF) fails to accurately reconstruct the vertical extent in all transects. Instead, the reflectivity is uniform along the cloud column. We see a continuous cloud signal between 5–15 km (Fig. 5, c). Contrasting, the Res-UNet predicts the vertical variability more precisely (Fig. 5, b). While the 2D profiles of the Res-UNet are smooth, the RF and OLS lead to a fragmented structure with a high value variability between the single pixels of the transect (I, IV). The examples show an inaccurate reconstruction of shallow clouds and multi-layer clouds for the OLS and RF.

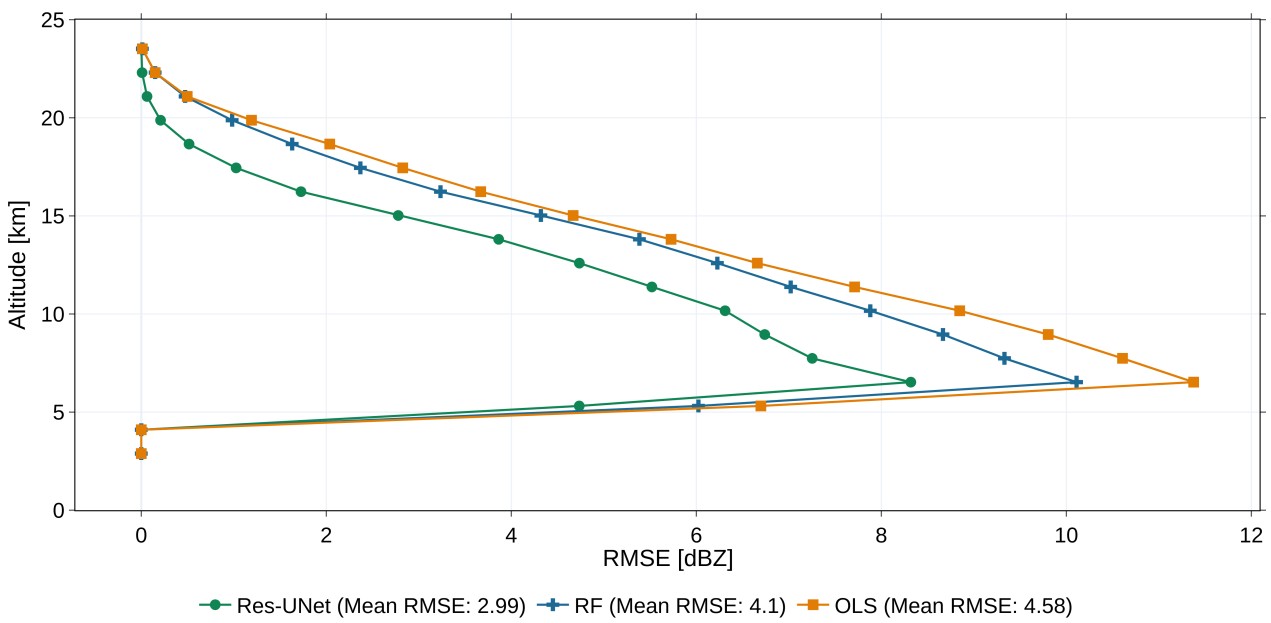

**Figure 4.** Height dependent RMSE for every height bin and the mean RMSE for all models calculated on the test dataset (n = 1500).

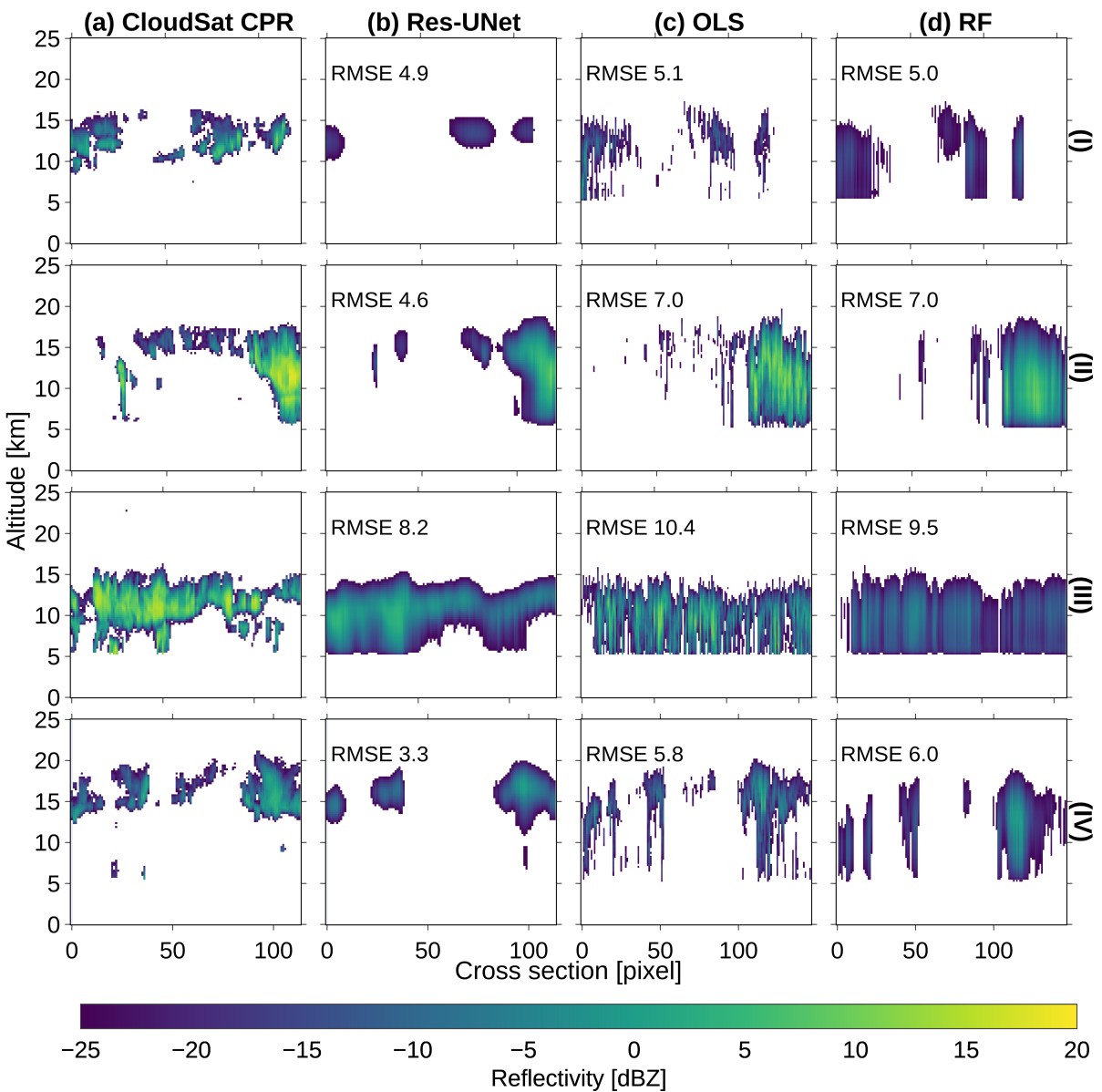

**Figure 5.** Reconstructing the radar cross section for four random examples of the test dataset (n = 1500). Values <= -25 dBZ are displayed transparent. We compare the reflectivity between the processed CloudSat cross sections **(a)** and the predictions of the Res-UNet **(b)**, the OLS **(c)**, and the RF **(d)** for each transect **(I)**–**(IV)**. The RMSE describes the error between the CloudSat data and the predicted profile.

### 3.3 Geographic analysis of the 3D cloud tomography

With the trained Res-UNet, we predict clouds on the MSG SEVIRI AOI. Since the network was trained using VIS channels, we cannot provide an accurate representation of the nocturnal cloud field. An exemplary 3D cloud tomography is predicted for 06 May 2016, 13:00 UTC. For that purpose, the satellite scene was divided into small subsets of overlapping 128 x 128 pixel images as described in Section 2.3.2. After feeding each subset into the network, the output tiles of 90 x 100 x 100 pixels were merged into a FD scene of 90 x 2400 x 2400 pixels for the whole AOI (Fig. 6, a).

The results contain a 3D cloud field along 90 height bins between 2.4–24 km. As shown in Figure 6 (a), the top-view on the maximum reflectivity per cloud column demonstrates the absence of hard borders between single prediction tiles. Even though CloudSat data is only available at the radar transects, we can extrapolate smooth cloud structures on the FD. The example tiles (b)–(d) point out a fluent transition between the edges of single prediction tiles (Fig. 6). Each example spans a horizontal extent of > 2.5 ° (100 x 100 pixels) to demonstrate the absence of artifacts between the tiles. High reaching convective complexes (b) and large-scale structures (c, d) are extrapolated at the FD scale regardless their location. Even though the overall reflectivity is underestimated, low-level and multi-layer clouds are displayed as contiguous entities.

We visualize the mean RMSE between 60 °N and 60 °S to investigate zonal variations for the test dataset. The geographic analysis is used to evaluate the reliability of the 3D cloud tomography. The RMSE shows a high latitudinal variability. At 30–50 °N, we observe the highest RMSE of 6–7 dBZ (Fig. 7, a). The RMSE at mid-latitudes on the southern hemisphere is lower than at the norther hemisphere. Nevertheless, the lowest RMSE is achieved in the tropics between 20 °N–20 °S. We analyze the RMSE in relation to the number of image-profile pairs originating the matching scheme in Section 2.1.3. Most image-profile pairs are located around the equator and the mid-latitudes (Fig. 7, b). Few pairs are matched around 10 °N and 30 °S. Regions in the mid-latitudes have the highest RMSE and the highest number of observations. In the tropics, the RMSE is lower. Here, we obtain a high amount of image-profile pairs from the matching scheme. The predicted cloud field points out a high geographic variability of the RMSE. We observe a higher RMSE for the northern than for the southern hemisphere. Clouds in the subtropics are more accurately represented than clouds in high latitudes. At the same time, we lack observations here. The analysis emphasizes the importance of the geographic location on the model predictions as well as the influence of the CloudSat orbit.

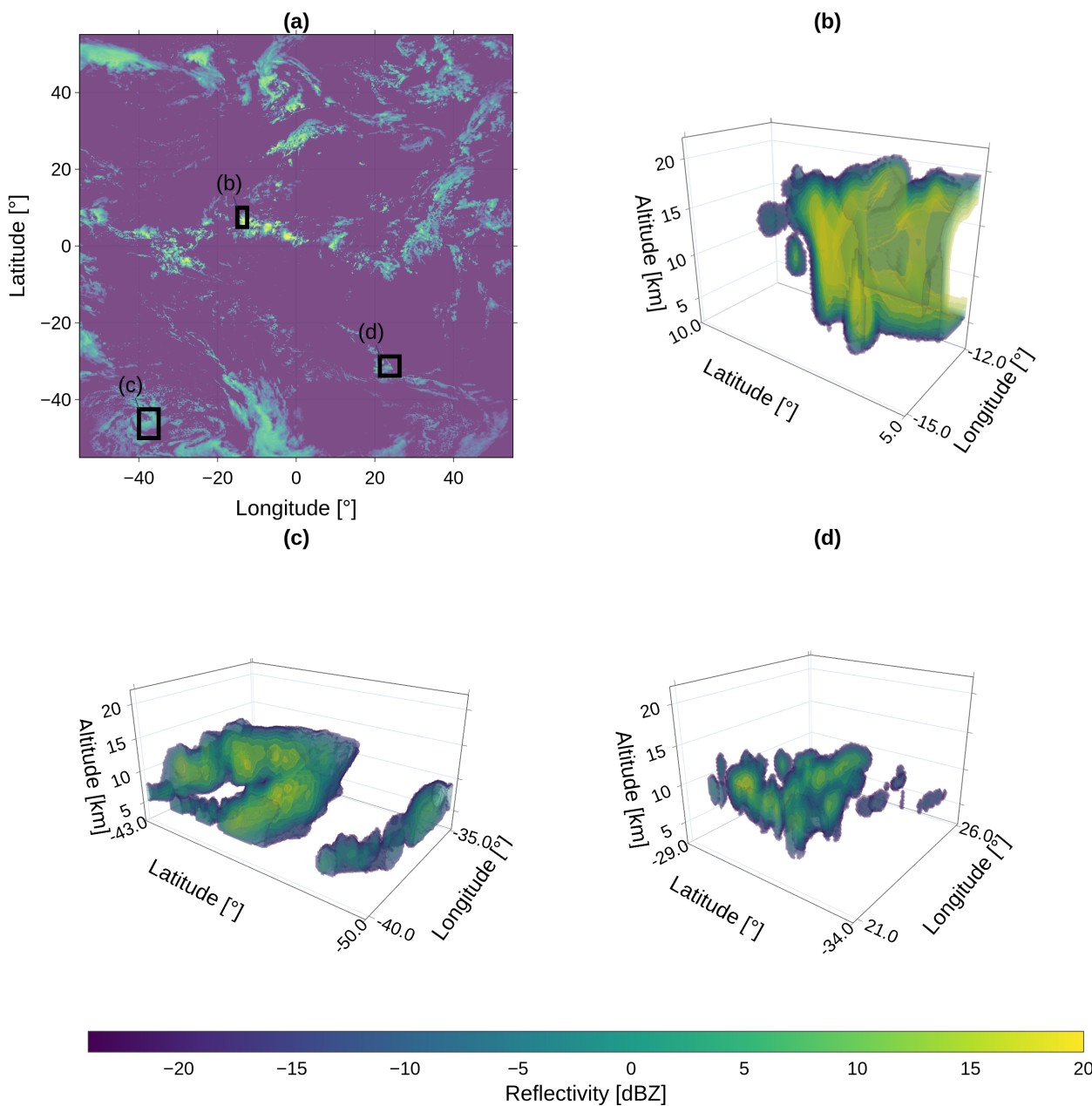

**Figure 6.** Prediction of 3D cloud structures from the Res-UNet along the FD MSG SEVIRI domain with a a top-view on the maximum cloud column reflectivity for each pixel on 06 May 2016, 13:00 UTC **(a)**. The detailed views in **(b)**, **(c)**, and **(d)** span several tiles of 100 x 100 pixels (2.5 ° on the geographic grid) to point out the absence of artifacts between predictions.

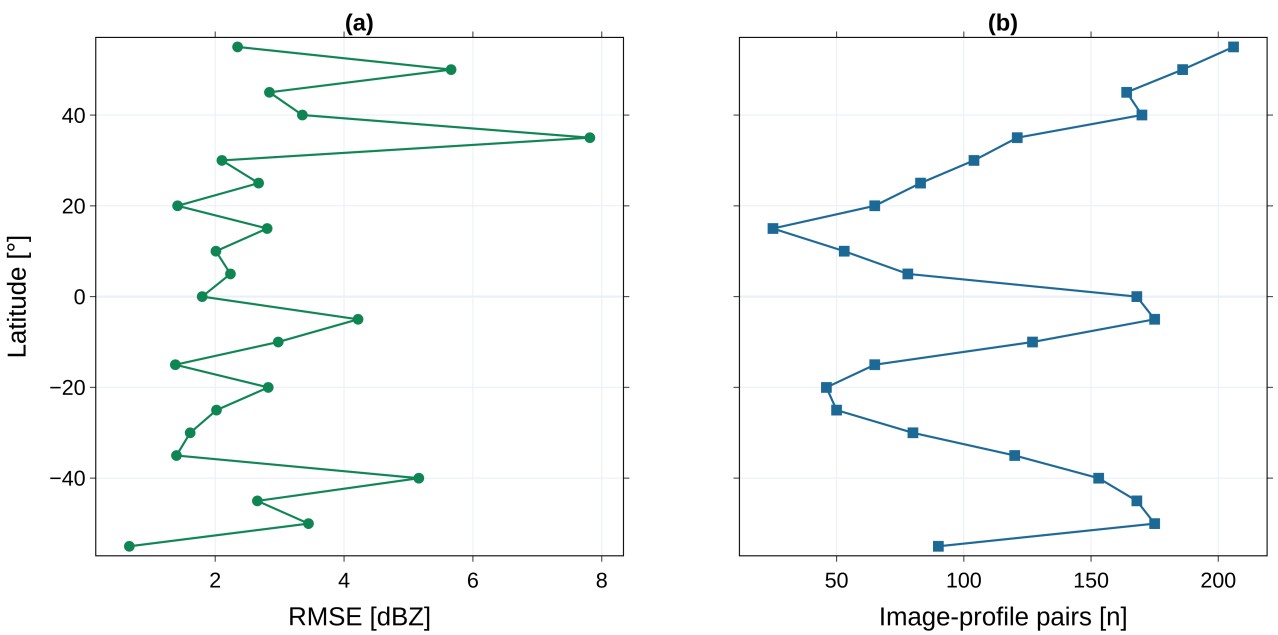

**Figure 7.** Zonal RMSE of the Res-UNet **(a)** and number of matched image-profile pairs used for model evaluation **(b)** for latitudes between 60 °N and 60 °S (n = 1500).

## 3.4 Comparison of the predicted CTH

To evaluate the quality of the Res-UNet predictions, we compare the reflectivity distribution between CloudSat and the Res-UNet predictions. In a second step, we calculate the CTH on the test dataset. The reflectivities in Figure 8 (a) are provided in logarithmic scale due to the high proportion of cloud-free pixels around -25 dBZ (Fig. 8 a, c). Values < -15 dBZ are visualized with a grey background. They lie below the threshold used to determine a cloud signal for the calculation of the CTH (Sect. 2.3.3). The distribution of CloudSat and predicted reflectivities is similar for up to -10 dBZ. For higher reflectivities, the distributions diverge. As demonstrated in Figure 2, the network fails to accurately reconstruct high reflectivities. The difference increases for reflectivities > 0 dBZ. Both reflectivity distributions are dominated by cloud-free pixels of -25 dBZ 8 (c). The comparison points out the importance on the background value for the whole distribution. For values > -15 dBZ, the difference between the distributions decreases. The shift of the distribution is reflected in the CTH in Figure 8 (b). Both datasets display a maximum CTH at 7 km height. This first peak is overestimated by the model. A second peak around 12–15 km height is underestimated by the Res-UNet. The difference between the predicted CTH is reflected within Figure 8 (d). The mismatch between the two peaks is about the same size. The underestimated second peak can be led back to the inaccuracy of the predicted reflectivities. The Res-UNet overestimates reflectivities < -15 dBZ at all height levels up to 15 km. It misses high reflectivities responsible for the peak of the CTH at 12–15 km (Fig. 3, b). Instead, we see an overall surplus of background values in the FD prediction.

Calculating the CTH on the FD predictions substantially increases the amount of available data points compared to the CloudSat data. Predicted images surpass the radar observations by a factor of 10000. We use the 3D cloud tomography to derive the FD CTH on the test dataset. For each time step, we calculate the CTH on the FD and aggregate the results to a monthly mean. These values are compared to the CLAAS-V002E1 product with a resolution of 0.05 ° (Finkensieper et al., 2020). The predicted CTH has a resolution of 0.03 °. Due to this mismatch, our predictions show more fragmented structures (Sect. 2.3.3). CloudSat faces sensor limitations in low and high altitudes of the troposphere (Sect. 2.1.2). While our analysis reveals an overall high agreement, the lack of e.g. thin clouds within the radar data can lead to a reduced similarity between the CLAAS-V002E1 data and the predicted CTH. We observe a connection between the similarity of the datasets and the hemisphere. In the northern hemisphere, the highest amount of image-profile pairs and the highest CTH difference occur between 0–20 °N. Between the tropics of the southern hemisphere, the amount of observations is similar whereas the CTH difference is considerably lower. The variability between the hemispheres can be led back to the distribution of land masses. A higher proportion of oceans in the southern hemisphere and a modified solar zenith angle affect the formation of clouds (Bruno et al., 2021). The result is an increased model performance which might be caused by the existence of either more uniform or less complex clouds.

Although the small-scale accuracy of the predicted CTH is improvable, the results allow an investigation of regional differences on the large scale. These differences arise especially around the equator and mid- to high-latitudes (Fig. 9). In mid-latitudes, the CTH over water bodies is overestimated in the southern hemisphere and underestimated in the northern hemisphere. These differences can be led back to an increased RMSE in these regions (Fig. 7). In contrast, a low RMSE in the

subtropics increases the accuracy of the predicted CTH. The model is biased toward predicting lower clouds than the observational data. Overall, the Res-UNet overestimates the occurrence of clouds in 6-8 km while underestimating high clouds (Fig. 8, b).

This issue is reflected within Figure 10. Here, we visualize the geographic distribution of the CTH difference (Fig. 10, a). The mean difference over all pixels accounts for 1.28 km. While the data show an overall agreement, the pixel-wise difference rises to a maximum of 10 km. This applies especially to regions in the subtropics. We observe an underestimation of the predicted CTH over land. Above the Atlantic ocean, especially in the tropics, the predictions are too high. The highest difference occurs in subtropics on the northern hemisphere (Fig. 10, b). At 20 °N, the mean difference accounts for 5 km. Around the tropics and mid-latitudes, both datasets are in higher agreement. The distribution of the CTH difference is inversely proportional to the amount of matched image-profile pairs (Fig. 7, b). The CTH difference decreases with an increasing amount of observational data from CloudSat. This applies to predictions over land and sea. Since we lack ground truth in the subtropics, the performance of the predictions decreases. The geographical differences are only partly in accordance to the distribution of the RMSE (Fig. 7, a). While the RMSE is lower in northern subtropics, the error of the predicted CTH reaches its peak (Fig. 10, b). The RMSE alone does not appear to be a suitable measure to define the reliability of the predicted reflectivity on the FD. This is due to influence of the skewed reflectivity distribution on the RMSE and its geographic variability.

Even though the comparison of the CTH points out regional differences, the predictions can be used to represent the CTH pattern on the FD. The CLAAS-V002E1 data is computed using the MSG SEVIRI imager data as well as derived products and additional data. Each of them bring their own bias, potentially multiplying their effects on the final CTH. In contrast, our CTH is based only on the predicted reflectivity. In that way, we can minimize the influence of additional data sources.

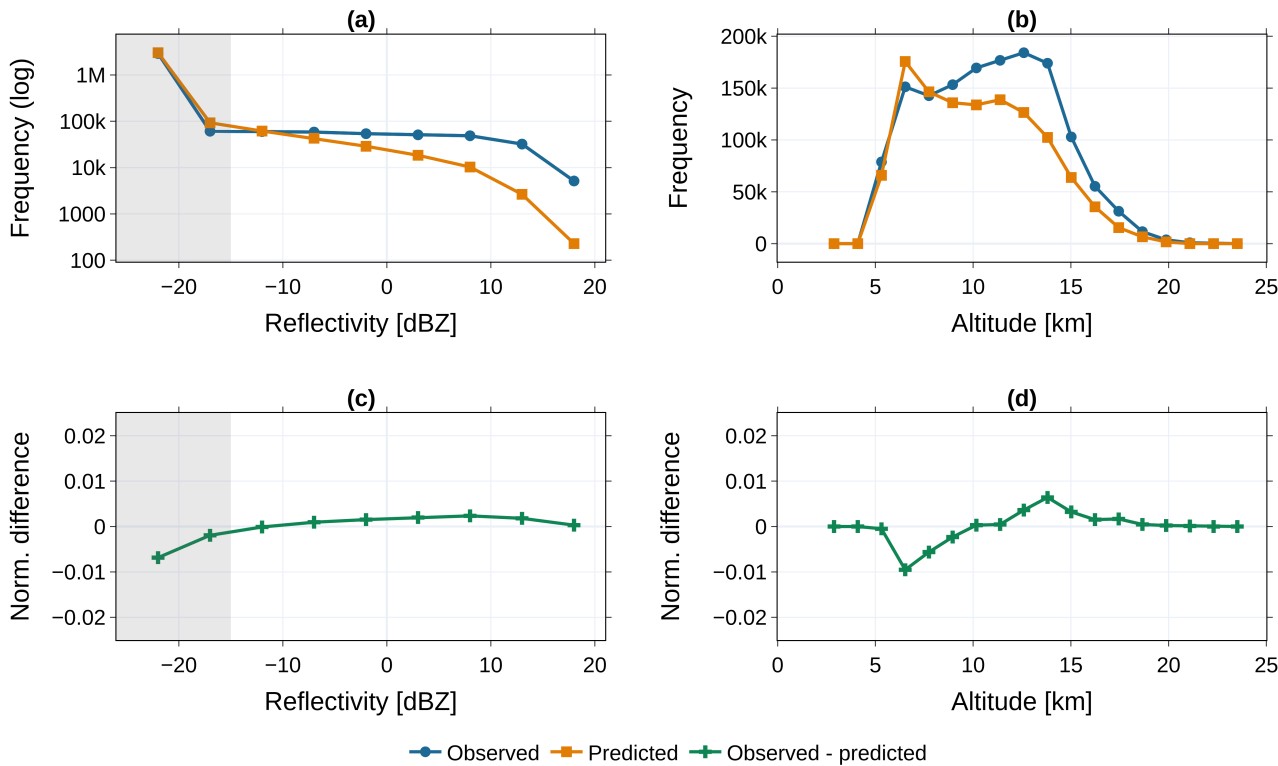

**Figure 8.** Comparing the reflectivity distribution and the derived CTH for CloudSat and the Res-UNet predictions. Data is calculated on the test dataset and aggregated to a monthly mean (n = 1500). The upper row frequencies **(a)** and **(b)** display the dBZ and computed CTH for observed and predicted data. Lower row images **(c)** and **(d)** show the difference between the observed and predicted data. Grey areas on the plots **(a)** and **(c)** contain reflectivities below the threshold of -15 dBZ applied for the CTH analysis.

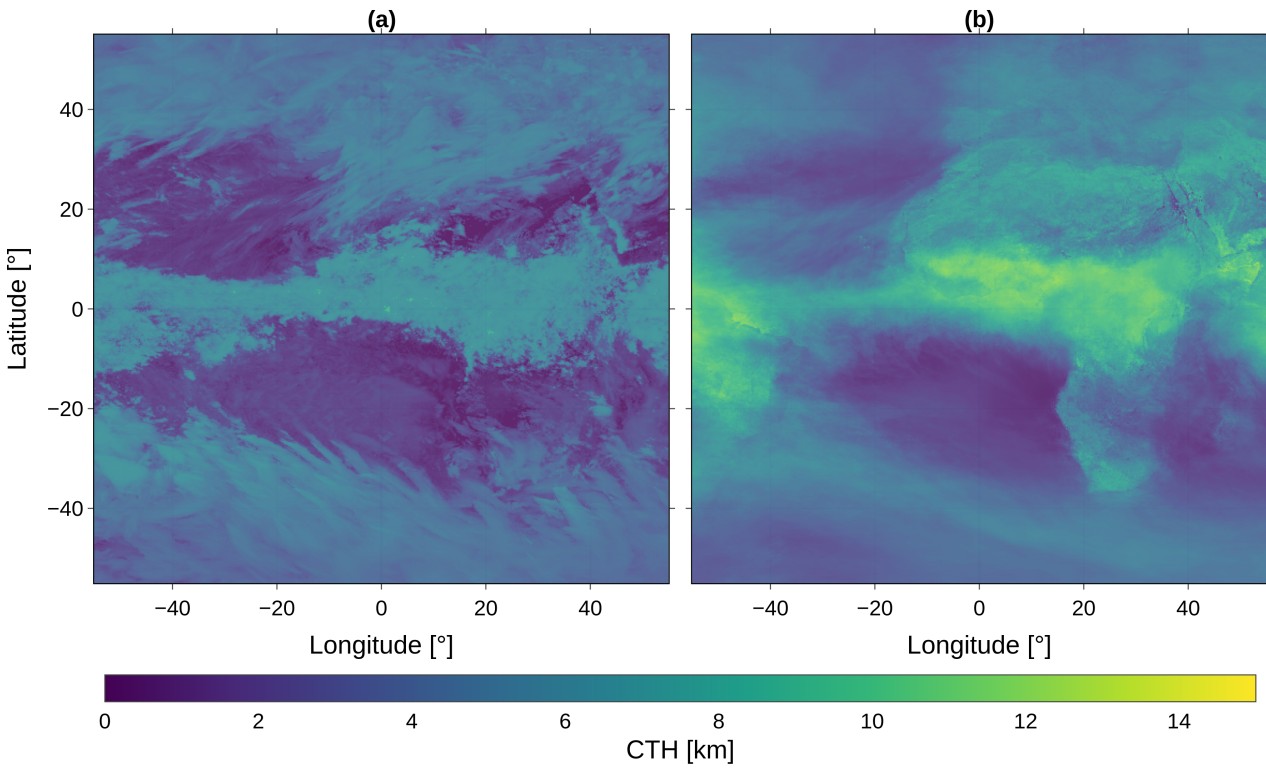

**Figure 9.** Monthly aggregation for the derived CTH for May 2016 **(a)** compared to CLAAS-V002E1 CTO **(b)**.

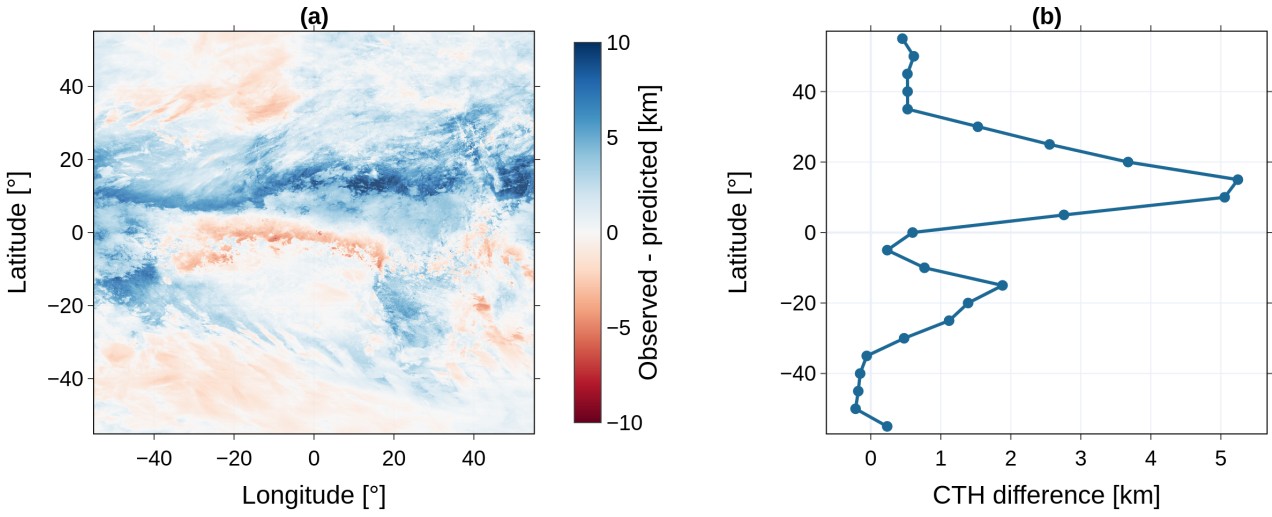

**Figure 10.** Difference between CLAAS-V002E1 CTO and the computed CTH for May 2016. Plot **(a)** shows the geographic distribution on the FD, **(b)** the zonal error.

# 4 Discussion

The Res-UNet makes predictions based on the MSG SEVIRI channels, preserving the spatial details and global context during training (Wang et al., 2022). The error of the model varies depending on cloud structure within the radar cross-section. Compared to pixel-based approaches like OLS, the Res-UNet better reconstructs the pixel connectivity. The OLS and RF operate on 1D cloud columns. This limits their ability to extrapolate cloud information to a larger scale, resulting in fragmented reconstructions (Fig. 5, c, d). While the RMSE and the reflectivity distribution are similar across all models, only the Res-UNet predicts a contiguous radar cross-section. Choosing a DL framework eliminates the need for prior predictor variable selection (Kühnlein et al., 2014; Leinonen et al., 2019). This can reduce the user bias in the input data (Jeppesen et al., 2019; Jiao et al., 2020).

The Res-UNet shows a 30 % improvement in the mean RMSE (Fig. 4). We could potentially further enhance the model performance by utilizing a more complex architecture. Our input data differs from typical gray-scale or RGB images, as it comprises multiple input channels and results in a 3D output (Drönner et al., 2018). Given the demands of our data and resource constraints, we adapted a standard UNet architecture rather than using a pre-trained model (Amato et al., 2020). Selecting the RMSE as a loss function can increase the blurriness in the results, particularly as model bias grows (Mathieu et al., 2016). This issue becomes apparent as all models struggle to predict high reflectivities (Fig. 2, b–d). The predictions are influenced by an imbalance within the CloudSat data, with the distribution of all models skewed toward low reflectivities. A resolution mismatch between CloudSat and MSG SEVIRI exacerbates this imbalance, causing peak reflectivities to blur out (Fig. 5, b–d).

Our study covers a large-scale AOI spanning 60 ° in all directions. In contrast to studies of Leinonen et al. (2019) or Hilburn et al. (2020), we incorporate a diverse landscape into our training. While Hilburn et al. (2020) focused on radar signal reconstruction over the USA using land-based radar, Leinonen et al. (2019) concentrated on radar cross-section prediction over the sea. In our study, we match image-profile pairs over land and sea to achieve model invariance to the topography. The performance of our Res-UNet is similar to their results. Nevertheless, we observe regional difference, especially for the CTH. We use a geographical analysis to highlight the importance of the topography and land-sea distribution and their impact on cloud microphysics (Wang et al., 2023).

We emphasize the influence of the geographic location and the CloudSat orbit, particularly in regions farther from the equator, where sensor accuracy diminishes (Fig. 7). Currently, we estimate the model error to be mainly influenced by the data imbalance and chosen loss function. In the future, addressing the cloud parallax shift in high-angle satellite observations could enhance the results by a more accurate image-profile matching (Bieliński, 2020). The most accurate predictions fall between 25 °N and 25 °S (Fig. 7), while mid- and high-latitudes exhibit a higher RMSE. This is likely due to the land-sea distribution and connected cloud patterns. Over ocean bodies, the model overestimates the reflectivity (Fig. 10, a). Using the water vapor channels could lead to this distortion. Improved predictions are evident over the southern hemisphere. Since CloudSat operates on a sun-synchronous orbit, it misses diurnal variations in each region (Sect. 2.1.2). In this study, we only derive daytime predictions. This is due to the influence of solar radiation in visible spectrum (VIS) channels (Hilburn et al., 2020; Jeppesen

et al., 2019). Additional distortions may arise from VIS channels, as imager data only represents the uppermost cloud layer. Depending on the location, they can be highly influenced by the surface albedo (Drönner et al., 2018). Training a model without the VIS channels can help to achieve predictions independent of the daytime. Reducing the extent of the AOI can mitigate the geographic performance differences but limits the applicability of the network. Training regional models and adjusting the loss function and model architectures offer potential solutions to improve the results of the 3D cloud tomography.

The reconstructed CloudSat cross sections are comparable to results achieved by Leinonen et al. (2019). For both studies, the RMSE varies between 0–1 dBZ for cloud-free samples, 3–7 dBZ for more uniform clouds, and more than 10 dBZ for multi-layer clouds. A common limitation is accurately representing multi-layer clouds. Using the satellite channels to derive these information may be limited (Schmetz et al., 2002; Thies and Bendix, 2011). High reflectivities tend to be underestimated due to noise near the ground (Stephens et al., 2008). To mitigate this, we exclude affected height levels, but this results in incomplete model predictions between 0—5 km (Fig. 2). Reducing noise is crucial for improving the performance of DL applications in remote sensing (Enitan and Ilesanmi, 2021). Our results are significantly influenced by the resolution difference between CloudSat and MSG SEVIRI, as well as the choice of the loss function (Sect. 2.1.3). The aggregation of CloudSat pixels blurs contrast within individual clouds (Fig. 5, a), which is further reflected in the increased RMSE. In contrast, Leinonen et al. (2019) use data from the MODIS satellite. It has a higher spatial resolution than the MSG SEVIRI data allowing for sharper predictions along the radar transect. However, polar-orbiting satellites like MODIS lack the spatio-temporal coverage of geostationary satellites (Dubovik et al., 2021). In their study, Wang et al. (2023) derive 24000 training samples for matching CloudSat and MODIS over six years. By using MSG SEVIRI data, we amplify the volume of training data. We have extracted approximately 30000 training samples from one year of imager data, which results in a ratio of about 1:7.

Currently, a compromise on the resolution is necessary to obtain predictions for Europe and Africa. However, promising new instruments are emerging. While data from comparable sources like the GOES-R series and the Himawari 8/9 satellites already offer a 1 km resolution, the recently launched Meteosat Third Generation satellite by EUMETSAT allows to close the gap and enables a more precise representation of individual clouds (Holmlund et al., 2021). Although our approach currently focuses on a region centered around 0 ° longitude, we can apply the same framework to other geostationary satellites, potentially achieving global 3D cloud coverage throughout the troposphere. The predicted cloud field can be valuable for time series analysis, enabling the tracking of clouds in four dimensions across space and time. Our results facilitate the identification of large-scale cloud patterns. They offer various applications, such as analyzing cloud organizational structures, pinpointing lightning locations, or conducting precipitation onset analyses. While we use CloudSat radar data as our ground truth, it needs to be evaluated whether this approach can be adapted to other 2D transect data sources, such as aerosol measurements. Former studies already derived aerosol properties from imager data (Carrer et al., 2010). The DL framework could help to achieve a full 3D retrieval of aerosols.

## 5    Conclusions

With the help of a neural network, we demonstrate for the first time the potential to infer comprehensive 3D radar reflectivities from 2D geostationary satellite images. While former studies were confined to a smaller region or the reconstruction of the 2D radar transect, we provide a framework to model the 3D cloud field in a high spatio-temporal resolution. The study is focused over Africa and Europe, but the approach can be used to predict the radar reflectivity on a global scale. Using only the predicted reflectivity, we derive the CTH without external data sources. Overall, the approach accurately reconstructs cloud structures in varying environmental conditions on the FD. Although the results are affected by sensor-specific and technical limitations, a vast potential for applications in atmospheric and climate sciences is apparent. With steadily growing data and the emergence of improved instruments, the results can close the consisting global data gap. We emphasize the benefit to extrapolate a 3D cloud field, especially in remote oceanic regions. Future work will focus on extending the proposed network by data with an enhanced spatial and temporal resolution and investigating 3D cloud processes in proceeding applications.

# Appendix A:  Overview of Res-UNet parameters

**Table A1.** Hyperparameters and training parameters of the Res-UNet.

| Type | Parameter | Value |
|---|---|---|
| | Depth | 4 |
| | Input channels | 11 |
| | Output channels | 90 |
| Hyperparameters | Filter size | 3 x 3 |
| | Pooling size | 2 x 2 |
| | Dropout | 0 |
| | Activation function | ReLU |
| | Number of epochs | 50 |
| | Batch size | 4 |
| | Input size | 128 x 128 |
| | Crop size | 100 x 100 |
| | Initial learning rate | 0.001 |
| Training parameters | LR scheduler (factor) | 0.1 |
| | Optimizer | ADAM |
| | Weight decay | 0.00001 |
| | Loss function | RMSE |
| | Augmentation (horizontal flip) | Randomness = 50 % |

# Appendix B:  Summary of trainable model parameters

**Table B1.** Total number of the Res-UNet model parameters.

| Total number of trainable parameters | Estimated total size (MB) |
|---|---|
| 1.893.328 | 194.27 |

*Code and data availability.* The source code for the imager data matching scheme and model framework are available upon request to the corresponding author and will be published with acceptance. Meteosat SEVIRI imager data used in this study have been downloaded at https://navigator.eumetsat.int/product/EO-:EUM:DAT:MSG:HRSEVIRI (EUMETSAT Data Services, 2023). The level 2B-GEOPROF CloudSat data have been downloaded at http://-www.cloudsat.cira.colostate.edu/ (CloudSat Data Processing Center, 2023). The CLAAS-

2.1 data were obtained from https://doi.org/10.56-76/EUM_SAF_CM/CLAAS/V002 (Finkensieper et al., 2020).

*Author contributions.* S.B and H.T. designed the study. S.B and S.N. developed the model code. S.B performed the modeling and visualization. S.B. and H.T. contributed to the model validation and analysis of cloud properties. S.B. and H.T. wrote the draft of the paper. All authors have read and agreed to the published version of the manuscript.

*Competing interests.* The authors declare that they have no conflict of interest.

*Acknowledgements.* The study is supported by the project "Big Data in Atmospheric Physics (BINARY)", funded by the Carl Zeiss Foundation (grant P2018-02-003). We acknowledge the infrastructure provided by the Max Planck Graduate Center Mainz. We acknowledge EUMETSAT for providing access to the Meteosat SEVIRI imager data. We acknowledge the Cooperative Institute for Research in the Atmosphere, CSU, for providing access to the CloudSat 2B-GEOPROF data. We acknowledge CM SAF for providing access to the CLAAS-2.1 data. We thank P. Spichtinger for useful discussions and comments on the manuscript.

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
