# Peer review of "AI-derived 3D cloud tomography from geostationary 2D satellite data"

_EGUsphere, 2023_

## Author Comment (AC1)

Thank you for taking your time to write the detailed review.

**Comment**: The most interesting part of the paper to me is the network structure that generates the 3D fields. The U-Net structure is appropriate for the problem at hand, although it is not quite clear from the description how the transformation from 2D to 3D is performed.

**Answer**: The transformation process from 2D to 3D is illustrated in Figure 1, Section 2.1.3, and Section 2.2. The matching algorithm stacks the 11 MSG SEVIRI channels of one sample together to one 3D image of 11 x 128 x 128 [C x H x W] pixels. Within this sample, CloudSat crosses a horizontal transect. The location of the cross section is defined as described in lines 104-109. We aggregate the CloudSat pixels on this track by their maximum value to fit the resolution of MSG SEVIRI. As a result, we receive a 3D image of 90 x 128 x 128 [Z x H x W] pixels containing the reflectivities along the cross section. After the downsampling, the transect has a width of one pixel in the 3D image (Fig. 1.3). Pixels outside the transect are assigned as missing values. In that way, we can preserve the location of the CloudSat data during training and use this information to extract the pixels from the 3D predictions to calculate the loss. The loss function is the RMSE (lines 168-171). As we are restricted to calculate the loss only for predictions along the transect, the loss reflects the model performance on only a small subset of the data (defining it as a sparsely supervised network). Despite these limitations, the final output consists of a full 3D prediction of each tile. Each tile contains a contiguous cloud field and is visually analyzed in Section 3.3 and Figure 5. We add an extended explanation in the revised manuscript.

**Comment**: While the network is interesting, I wonder about the usefulness of the model outputs since the model clearly suffers from regression to the mean and blurriness of the results. Also, the uncertainty of the outputs is not estimated. Could you comment a bit on which applications would find the current results useful? And how the above mentioned issues could be improved?

**Answer**: As mentioned in lines 242-243 and 305-307, we agree that the network suffers from problems connected to the regression to the mean and the blurriness of results. In the revised manuscript, we will address these problems and possible improvements more straightforward.

The histogram of the CloudSat data shows a shift towards the background value of -25 dBZ (Fig. 6). To address this imbalance, and, to enhance the model performance, we use the UNet with additional residual connections (Res-UNet) and an automatic learning-rate optimizer. Using a full year of satellite data leads to the generation of approximately 30.000 samples. We reduce the percentage of cloud-free samples face the imbalance between cloudy and clear-sky samples (lines 116-119). Still, a cloudy sample can have a substantial proportion of background values. As reducing the size of the dataset affects the model performance, we randomly flip the samples during training for data augmentation. The goal is to include model invariance to the cloud orientation in the model and avoid overfitting. Although we use techniques to deal with the regression to the mean, the results show room for improvement. All predictions smear out. This is not caused only by the imbalance of the data, but can be led back to the loss function. The RMSE is known to enhance the blurriness of the predictions. An adapted loss may improve the results. Nevertheless, the study suffers from the resolution mismatch between the MSG SEVIRI and the CloudSat data. Aggregating the CloudSat pixels further increases the blurriness of the ground truth as well as those of the predictions (lines 109-112). An improvement can be achieved by using a deeper architecture along satellite data with a higher resolution (e.g., GOES-R, Himawari 8/9, or, in the future, MTG). As stated in lines 318 – 321, the goal of our study is to point out for the first time the possibility to derive continuous 3D clouds using DL. We see the innovation of our work not in the network architecture itself as it was kept simple on purpose. Nevertheless, using powerful architectures like the Vision Transformer as well as an adapted loss can improve the above mentioned issues.

Despite current performance issues, we see a vast potential for the method in applications of atmospheric and climate sciences. In Section 3.3, we exemplary compute the CTH to evaluate the quality of the predictions for analyzing cloud microphysics (lines 255ff.). Further use-cases comprise e.g., the analysis of cloud organizational patterns, the identification of lightning locations, or a detailed analysis of precipitation

onset. We use the CloudSat radar data as ground truth, but the approach can be transferred to different data on a 2D transect, e.g., aerosol measurements. The absence of artifacts between predicted tiles on the FD enables an investigation of 4D movement patterns through space and time (lines 249-254). The data can be used together with simulation results to improve the representation of large-scale cloud dynamics.

In the next section, we shortly address your specific comments that will be added in the revised manuscript.

- lines 24-25:

  - **Comment**: "Passive sensors such as geostationary satellites": the statement needs more precision, satellites are not sensors

  - **Answer**: Changed to "While observations from passive sensors on geostationary satellites are limited to monitor the uppermost atmospheric layer from space, [..]".

- lines 36-37:

  - **Comment**: "The large-scale generability of these methods is expandable since their 3D results are limited to the cloud's spatial vicinity": I don't understand this sentence, please clarify.

  - **Answer**: We agree that this sentence might be unclear. We will omit it as it is not necessary for the central statement of the paragraph. The relevant information can be found in the sentences before and after this one.

- line 48:

  - **Comment**: "time efficiency and feasibility": it's not clear to me what this means

  - **Answer**: DL networks do not require feature extraction like traditional ML methods. That reduces the time necessary to start the training and the user generated bias. Also, their performance on big data is better. We change the wording to avoid misunderstandings.

- line 80:

  - **Comment**: "orbiting the globe on a sinusoidal track": how does a satellite orbit on a "sinusoidal track"?

  - **Answer**: This refers to the radar track visualization in 2D. Changed to: "The ground truth of the study is derived from an active radar on board the CloudSat satellite which moves on a sun-synchronous orbit."

- lines 91-92:

  - **Comment**: "resampled to a geographic grid": what kind of grid, a lat-lon one?

  - **Answer**: Yes, data is resampled to a spatial grid in geographic coordinates with a resolution of 0.03° (lines 90-92). The missing information will be added.

- Section 2.1.2:

  - **Comment**: CloudSat is on a sun-synchronous orbit, meaning it sees every location at the same local solar time. This might introduce some diurnal bias to the data; this should be acknowledged.

  - **Answer**: We appreciate that remark and add following: "Since CloudSat follows a sun-synchronous orbit, it receives information on the cloud reflectivity at different locations along the globe always at the same local time. This reduces its ability to reflect the diurnal variability within each region of the AOI.".

- line 109:

  - **Comment**: The use of "XY" is confusing, I read this initially as "X times Y" but apparently you mean a diagonal transect through the image? Or did I misunderstand?

  - **Answer**: "XY" refers to the diagonal transect of the CloudSat radar along the pixel dimensions X and Y of the MSG SEVIRI image. We change the naming to "transect" or "cross section" in the text (in figures as [Z,(H,W)]).

- line 125:

  - **Comment**: "smoothing" should probably be "filtering"

  - **Answer**: We agree and change to "filtering".

- line 137:

  - **Comment**: The use of the word "delineate" here and a couple of other places in the places seems incorrect

  - **Answer**: Changed "delineate" to "predict" (line 7, line 137) or "estimate" (line 81).

- line 159:

  - **Comment**: I would like some more details on how the network structure maps the 2D input fields to the 3D output fields. Is the channels dimension used transformed to the Z dimension of the output?

  - **Answer**: See Figure 1 and Section 2.2. The MSG SEVIRI data consists of a 2D field for each satellite channel. During the matching scheme, these channels are stacked to one 3D image of 11 x 128 x 128 pixels. On the encoder side of the network, the channel dimension of the MSG SEVIRI input is expanded to the proposed filter size of 256. On the decoder side, these filters are reduced and mapped to a model output size of 90 x 128 x 128 pixels. As a result, the channel dimension of the output equals the Z dimension of the CloudSat ground truth. We will add an extended explanation in the revised manuscript.

- lines 163-164:

  - **Comment**: How is the 3D scene predicted by the network compared to the CloudSat data during training? CloudSat only gets a 2D vertical cross section of the scene. Is only part of the scene selected for comparison? Also, what loss function do you use for training?

  - **Answer**: Seems to be similar to the question dealing with the transformation from 2D to 3D in the upper part of the text. In summary, we transform the 2D cross section of CloudSat to a 3D image of 90 x 128 x 128 pixels during the matching scheme. This 3D image has the same size as the model output after training. For the CloudSat 3D image, only pixels along the transect hold finite values. These can be used to filter both images and calculate the RMSE between the observed and predicted cross section (sparse supervised network). We will add this information in the revised manuscript.

- line 176:

  - **Comment**: "Both models": unclear which models this refers to

  - **Answer**: Refers to the pixel-based methods mentioned in line 172. We rewrite the sentence for clarity.

- line 181:

  - **Comment**: "pictures" is used incorrectly here.

- **Answer**: Changed to "The RF is a supervised ML algorithm which provides robust results when working with environmental datasets in the natural sciences [...].".

- lines 189-190:

  - **Comment**: In the joined 2400 x 2400 pixel 3D prediction, is the field continuous at the borders of the 100 x 100 pixel tiles? Or do you see discontinuities or artifacts?

  - **Answer**: The joined 3D prediction consists of a continuous field without artifacts. As shown in Figure 5, each of the sub-figures (b) – (d) represents a combination of several 100 x 100 pixel tiles ( 2.5° on the Lat-Lon grid). The results are described in Section 3.3, lines 250 – 252 and will be explained in more detail in the revised manuscript.

- line 220:

  - **Comment**: "That said" seems out of place here - please revise.

  - **Answer**: Changed to: "[...] The DL network indicates an underestimation of high reflectivities and an overestimation of low reflectivities for low-level clouds.".

- line 231:

  - **Comment**: I don't understand that "denominational structure" means here.

  - **Answer**: Refers to the results for the OLS and RF in Figure 4. In contrast to the DL results, the reconstructed cross section is not as smooth but shows a rather fragmented structure.

- lines 262-263:

  - **Comment**: High, thin ice clouds may also not be observed by CloudSat due to being under the minimum detectable reflectivity.

  - **Answer**: Thank you that remark. We will change the text in the revised version.

- Figure 7:

  - **Comment**: Maybe you could add a panel showing the difference of a and b to illustrate the biases better.

  - **Answer**: Interesting suggestion, we add that figure in the revised manuscript.

- line 288:

  - **Comment**: "Leaving out the affected channels downgrades the overall performance": it would be good to see something to demonstrate this.

  - **Answer**: We tried a model setup without VIS. Due to its poor results, we did not include it in the manuscript for brevity. In the revised manuscript, we will rewrite the text in this paragraph.

- lines 292-293:

  - **Comment**: "In contrast to pixel-based DL methods like the CNN or CGAN, the Res-UNet utilizes a larger receptive field preserving the spatial dimensionality and global context information during the training routine." This is not a correct statement regarding the CNN or CGAN architectures. CNN architectures can also achieve large receptive fields and global context using downsampling. In fact the UNet itself is a type of CNN - its distinguishing feature is the addition of skip connections to preserve resolution. As for the CGAN, it refers to a certain training setup of generative models that could be implemented with either normal CNNs or with (Res-)UNets.

- **Answer**: We agree and rewrite the sentence: "Like other CNN architectures, the Res-UNet preserves the spatial dimensionality and global context information during the training routine (Wang et al., 2022). Compared to pixel-based methods like the OLS, it reconstructs the spatial connectivity between the pixels more accurately and enhances the representation of the continuous cloud field. [...]".

- lines 312-313:

  - **Comment**: Approximately 1 km resolution is also already available from the GOES-R series and Himawari 8/9 satellites.

  - **Answer**: We know about the resolution of these satellites, but we decided to stick with the MSG satellite to fit our main study area. For a potential global composite of the 3D predictions, data from GOES-R and Himawari 8/9 may help to improve the results. We add this information and change lines 311-314: "At the moment, a compromise on the resolution is necessary to obtain predictions centered over Europe and Africa. However, newly emerging instruments offer an enticing prospect to tackle this information loss. While comparable data sources like the GOES-R series or the Himawari 8/9 satellites already provide data in a resolution of 1 km, the recently launched satellite Meteosat Third Generation by EUMETSAT (Holmlund et al., 2021) is expected to close the data gap. Together, they can be leveraged to investigate a 4D reflectivity field through space and time".

- lines 321-322:

  - **Comment**: "Since it is independent of external or interconnected data sources, the bias within the data is reduced.": unclear sentence, I'm not sure how the latter follows from the former.

  - **Answer**: The operational CM SAF CTH is computed using the MSG SEVIRI satellite data as well as derived products and additional data. Each of them may bring their own bias, potentially multiplying their effects on the final CTH. In contrast, our CTH is based only on the predicted reflectivity. In that way, we can minimize the influence of additional data sources. We will rewrite this explanation in the revised manuscript.

---

## Author Comment (AC2)

**General comments:**

- **Comment**: In my opinion, any application of machine learning to scientific research questions requires some level of uncertainty quantification. I appreciate that this might go beyond the scope of the paper, but estimating model uncertainties would greatly improve the trustworthiness of the results, and likely help inform how (and for which scenarios) the model needs improving.

- **Answer**: We agree this is an interesting idea. We did not include a quantification of model uncertainties since it seemed out of scope of our initial submission. We will evaluate whether and how to integrate a form of uncertainty quantification in the revised manuscript.

- **Comment**: The writing is at times too verbose and unclear. I am including a number of specific comments below, but I would suggest going through the paper again and streamlining the narrative. At times, the section headings could be more specific. Some of them (e.g. "Comprehensive predictions") are not particularly descriptive of the work done.

- **Answer**: We revise the manuscript with particular attention to this issue.

- **Comment**: Please add tables of the training and hyperparameters in the paper (or better Supplementary Information). Most importantly, please include information about the number of parameters of each of your models.

- **Answer**: The tables will be added in the revised manuscript.

**Specific comments:**

- Line 9:

  - **Comment**: Abstract: "average error" is a bit vague and not the same as RMSE. Please revise. Furthermore, it is not clear what "total value range" in this sentence is referring to.

  - **Answer**: Changed to: "The RMSE after training is 3.41 dBZ which equals a difference of 7.5 % compared to a reflectivity scale between -25 and 20 dBZ."

- Line 27:

  - **Comment**: "[...] spatially and temporally limited perspective [...]." Please expand on what the spatial and temporal characteristics of the different instruments used in this work are. Specifically, I think the authors could expand on the benefit of geostationary satellites when it comes to temporal resolution, as polar-orbiting satellites often require 1+ days to observe the same area again.

  - **Answer**: The spatial and temporal characteristics of the specific instruments are described in Section 2.1.1 and 2.1.2. We agree that some points may be missing. In the revised manuscript, we add a more detailed comparison.

- Line 29:

  - **Comment**: "While this analysis often rests upon subjective labeling or fixed thresholds [...]." Please add example citations to previous work.

  - **Answer**: We change the text adding examples for the retrieval of the cloud optical depth (https://doi.org/10.1175/2011JAMC2601.1) and the cloud effective radius (https://doi.org/10.5194/acp-20-1131-2020).

- Line 33:

  - **Comment**: "joined" ==> joint

- **Answer**: Changed to: "A combined use of different instruments to derive comprehensive 3D structures [...]".

- Line 36:

  - **Comment**: I am not sure what "generability" means.

  - **Answer**: This sentence is unclear, we omit it. The sentence before and after (lines 35, 37) are sufficient for our statement (that, to our best knowledge, no large-scale 3D interpolation of the cloud reflectivity from radar exists).

- Line 54:

  - **Comment**: I would suggest including other works, such as https://arxiv.org/abs/1911.04227, when discussing cloud classification.

  - **Answer**: We appreciate that remark and revise the citations.

- Line 70+:

  - **Comment**: The sentence is very long and hard to follow. Please rephrase.

  - **Answer**: Changed to: "Former studies had a focus on reconstructing the 1D or 2D cloud column. In contrast, we apply a DL framework to predict the radar reflectivity not only along the radar cross section, but on the satellite full disk (FD)."

- Line 78:

  - **Comment**: "[...] originates a geostationary satellite." ==> "from" missing.

  - **Answer**: Changed to: "The input data for the network originates from a geostationary satellite [...]".

- Line 84:

  - **Comment**: The EUMETSAT abbreviation was already used in line 78, before it was defined here.

  - **Answer**: Moved the definition to line 78.

- Figure 1:

  - **Comment**: It would be great if you could add information about the size of the boxes shown in Part 1, to make it more clear how the matching algorithm works. Furthermore, it was confusing to see 90 height levels in the figure, when the text previously described CloudSat to have 125 height levels. Could you also comment on the missing data shown in the CloudSat profile?

  - **Answer**: We revise the figure adding an extended description of the matching algorithm. The boxes have a size of 128 x 128 pixels. The missing data in the profile represents every pixel with a value of -25 dBZ (lines 125-127). In Figure 1, these values are transparent to highlight the profile itself. We add this explanation in the figure caption. The reduction of the height levels is described in lines 123-124, but we restructure the section for clarity.

- Line 114:

  - **Comment**: "Extracted satellite samples display the physical predictors fed into the network [...]." Please rephrase, as it is not clear to me what you mean.

  - **Answer**: Changed to: "The matching algorithm extracts the image-profile pairs. We use the information of the satellite channels within these pairs to reconstruct the vertical cloud distribution."

- Line 115:
  - **Comment**: I would rename "samples" as "matched image-profile pairs" or similar to be more exact.
  - **Answer**: We appreciate that suggestion and revise the phrase.
- Line 117:
  - **Comment**: How come you have no test set? Validation sets are great, but susceptible to "human gradient descent", since we usually justify model modifications by improved performances on the validation set. This doesn't necessarily mean that the model is better, it just means that performance on these specific examples is improved.
  - **Answer**: The model is trained and validated on data from 2017 (lines 115-116). We use nine months for training (here January-September) and three months for validation (October-December). Due to limited resources, data from May 2016 (n=1500) is used as a test set to evaluate the model performance and to calculate the CTH. We will revise the text and the reported figures to ensure they represent the performance on the test set. In this study, we apply a standard architecture with slightly modified learning-rate and weight decay. In the manuscript, we add a more detailed training protocol.
- Line 123:
  - **Comment**: How was the reduction from 125 to 90 height levels done?
  - **Answer**: We analyzed the CloudSat quality flag for the radar profiles to identify height levels with a high proportion of noise (lines 124-127). The original radar scene contains a high amount of noisy pixels up to 2-3km height. That is why we crop off the lower 10 height levels. The reduction affects the representation of low clouds and the cloud base, reducing the model performance in these altitudes. We find an additional cloud free region $> 20 - 30$ km (https://doi.org/10.1029/2008JD009982, line 19). To reduce the proportion of cloud-free pixels, we cropped these upper height levels off. The final Z-dimension consists of 90 height levels (10 - 100, between 2.4 km and 24 km). We add an extended explanation in the revised manuscript.
- Line 140:
  - **Comment**: "By seeking non-linear approximations of between the input and the output data [...]." Please rephrase.
  - **Answer**: Move the sentence up to line 134 and changed to: "Neural networks have the potential to capture highly complex relationships between input and output data. The Res-UNet displays a modified framework designed for the use-case of remote sensing data. [...]"
- Line 163:
  - **Comment**: "As flipped images are perceived as new samples, we enhance the amount of training data by giving all samples a chance of 25% to be either vertically or horizontally rotated." Please comment on whether these transformations are valid for the context of satellite measurements, especially CloudSat, with its ascending and descending orbits. Are the two acquisitions totally equal, in that flipping one can simulate the other?
  - **Answer**: The data augmentation used here is not meant to simulate ascending/descending orbits. Instead, by applying random flipping, we aim at including invariance to the cloud orientation in the model and avoid overfitting. A structured analysis of differences between ascending/descending CloudSat orbits is out of scope for this paper. Since data from both

orbits are included in the training set we don't expect a different performance of the model on either orientation.

- Equation 3:
    - **Comment**: Either define all variables in the equation, or remove from the paper, as the RMSE is a relatively standard quantity.
    - **Answer**: We remove the equation in the revised manuscript.

- Line 173-174:
    - **Comment**: Rephrase "horizontal diagonal".
    - **Answer**: Changed to "radar transect" or "radar cross section".

- Figure 3:
    - **Comment**: Please explain how many samples the joint plot was calculated over.
    - **Answer**: The plot was calculated over all samples of the test set (n=1500). We add the number of samples in the revised figure.

- Line 218+:
    - **Comment**: This sentence is very hard to follow. Please re-write.
    - **Answer**: We revise this paragraph. This sentence will be changed to: "In the joint plot, values around 0 represent a high agreement between the observed and predicted distribution. On these height levels, the observed reflectivities are most accurately reconstructed. We observe areas of high agreement in shape of a curved line reaching from high to low altitudes for all of the three models."

- Line 220+:
    - **Comment**: Do you have any idea why you observe different performance trends for your pixel-based and Res-UNet approaches?
    - **Answer**: After revising, we think this was due to an error in the code. The results could differ as a result of the regression-to-the-mean. Since the DL network is fed a lot of nearly cloudfree images, they can lead to a shift of the distribution. In contrast, extreme values have a higher influence on the OLS and RF. This can affect the prediction of positive values. Nevertheless, we change the plot to all models pointing towards the same direction.

- Line 226:
    - **Comment**: How were the four samples chosen?
    - **Answer**: Those four samples were randomly chosen from the test set. We add the missing information in the revised manuscript. We add this information in the manuscript.

- Line 228:
    - **Comment**: What do you mean by "transferability" in this context?
    - **Answer**: Refers to the ability of the network to better represent the horizontal and vertical position of the clouds along the transect. Wee see this is unclear and omit the sentence, as the information is more precisely expressed in the sentences before and after.

- Line 231:
    - **Comment**: "A denominational structure [...]." I am not sure what you mean by this.

- **Answer**: Refers to the results for the OLS and RF in Figure 4. In contrast to the DL results, the reconstructed cross section is not as smooth but shows a rather fragmented structure.

- Figure 4:
  - **Comment**: Please label each subplot, and refer to the subplots as you discuss the results in the main text to make it easier to follow your arguments.
  - **Answer**: Thank you for the remark, we add this in the revised manuscript.

- Line 233:
  - **Comment**: "[...] the Res-UNet shows more robust results [...]." I am not 100% convinced that the RMSE and visual inspection of the results qualify this sentence. Have you looked at any other metrics, or quantitatively studied performance as a function of cloud type for more than a couple of samples?
  - **Answer**: We did not derive the cloud type as it was not within the scope of the study. But we used not only a few samples for the evaluation but we calculated the RMSE for all models on the test set (n=1500). Over all height levels, the RMSE for the Res-UNet decreases by 30-35% compared to the OLS & RF. This percentage is used to support our hypothesis.

- Line 235+:

  **Comment**: I am not sure I follow your discussion about quality flags. Please clarify.
  **Answer**: We refer to the internal CloudSat flag that describes the quality of the received reflectivity (lines 125-127). Reflectivity values are filtered by a minimum threshold of six and set to -25 dBz (our background value). As this affects almost all values in low altitudes, our network reconstructs only the cloud signal above 5 km height. We will revise this paragraph.

- Line 238-239:
  - **Comment**: "[...] this leads to a lower model uncertainty." Do you mean "lower error"? Model uncertainties to me means quantification of how certain a model is in its predictions.
  - **Answer**: Yes, this refers to the model error and the difference between the observed and predicted distribution. We change the manuscript accordingly.

- Line 239:
  - **Comment**: "That said [...]." This sentence isn't quite clear to me. Please rephrase/clarify.
  - **Answer**: Changed to: "Leaving out noisy pixels is needed to improve the model performance. At the same, this results in a loss of information in low altitudes. This issue is reflected within the results. [...]"

- Line 260:
  - **Comment**: "The first peak [...]." It would be really helpful if you could refer to the labels of the subplots when discussing the results.
  - **Answer**: We see this is confusing and revise the section to be more clear.

- Line 264+:
  - **Comment**: "These channels are identified as essential information [...]." It would be great if you could show evidence for this, or expand the discussion on what makes you draw this conclusion.

- **Answer**: This statement is based on a different model setup without VIS. In the initial submission, we did not add this for brevity, but we revise the paragraph and add supporting information.

- Line 267+:

  - **Comment**: "Comparing [...] reveals an overall high agreement." Do you have any quantitative comparisons between your model outputs and the CTH from CLAAS, or is this mainly from visual inspection?

  - **Answer**: The initial comparison was visual. In the revised manuscript, we add a quantitative analysis.

- Line 277:

  - **Comment**: "The approach offers [...]." Which approach are you referring to?

  - **Answer**: Refers to computing the CTH only by predicted reflectivities. We change that paragraph to be more precise.

- Figure 6:

  - **Comment**: Is the comparison calculated across the entire FD of the model predictions, or just the tracks that overlap with CloudSat? If it's non-normalized frequencies, they should be calculated across the same number of datapoints to be comparable, if I am not mistaken? Please clarify in the caption and main text.

  - **Answer**: Thank you for that remark, we will check the figure concerning the number of datapoints and revise the corresponding text.

- Discussion section:

  - **Comment**: I found it at times hard to follow when you are referring to your own work, versus work done by other people. Could you please go through this section again?

  - **Answer**: We revise the manuscript with particular attention to this issue.

- Line 280:

  - **Comment**: Please clarify what you mean by "minimal architecture".

  - **Answer**: This study is based on a small and rather standard architecture (UNet added by residual connections). We want to point out the possibility to interpolate continuous 3D clouds from the 2D data sources. That is why we stick to the UNet instead of using a more complex and powerful architecture (like ViT). We rewrite the sentence for clarity.

- Line 282:

  - **Comment**: "others" ==> others' work

  - **Answer**: Changed in the revised version.

- Line 284:

  - **Comment**: "Nonetheless, defining those variables as additional predictors has a negligible effect on the model performance." Are you showing evidence for this somewhere?

  - **Answer**: Similar explanation as for lines 264+. We tried a different model setup. Since it achieved poor results, we did not add it in the initial submission for brevity. In the revised manuscript, we change that paragraph.

- Line 288:

– **Comment**: "Leaving out the affected channels downgrades the overall performance." Same here, do you have evidence supporting this statement?

– **Answer**: Same explanation as above, will also be changed.

---

## Author Comment (AC3)

Thank you for taking your time to write the detailed review.

**Main points:**

- **Comment**: This looks to me as if this study has the characteristics of an ML-based retrieval, where to properties being retrieved from the SEVIRI radiances are the vertical profile of radar reflectivity. This has some similarities to other algorithms, such as the GPROF retrieval (Kummerow et al, J. App. Met., 2001), which could be discussed. Perhaps the method in this paper could produce improved precipitation retrievals in a similar style to the GPROF retrieval (although not in this publication)?

- **Answer**: That is an interesting suggestion worth to be investigated. We see the similarity to reflectivity-based methods for precipitation retrieval and thought about this aspect before. A detailed analysis seemed out of scope for this paper. Nevertheless, we think this would be an interesting follow-up.

- **Comment**: Although it doesn't seem to be at the level where I could be comfortable using the retrieved reflectivity profiles yet, but perhaps there are certain conditions where the data could be more trustworthy? Would limiting the study to a smaller area (e.g. close to nadir for SEVIRI) produce better results?

- **Answer**: We agree with you on this issue. In our paper, we state the current problems with the method and its limits in terms of accuracy (Section 4). Resampling the satellite data on a spatial grid with geographic coordinates affects the accuracy of the input data with greater distance to the equator (Section 2.1.1). We think limiting the area to e.g. the tropics has the potential to improve the results. In the revised manuscript, we add a visualization of the zonal error to display the geographic differences.

**Line by Line comments:**

L9 -**Comment**: Bias in retrievals based on approximations and cloud physical properties L9 - It is later stated that RMSE varies significantly by cloud situation - how is this value derived? **Answer**: We did not classify the cloud type, but we investigated the RMSE on the test set with 1500 samples. We rephrase that sentence for clarity.

L10 -

- **Comment**: receive high accordance -> find high agreement?

- **Answer**: Thank your that remark, will be changed.

L30 -

- **Comment**: at risk of bias. I would argue this is almost always an issue (and would be for a ML method too)

- **Answer**: You are right, no method is probably completely free of bias. We will rephrase that sentence.

L30 -

- **Comment**: If passive sensors lack this information, how can it be recovered from this method?

– **Answer**: Information on the cloud column below the cloud top can be approximated using the satellite channels at different wavelengths (lines 27-29). While they enable an approximation, their information density is lower compared to the radar reflectivity received from active radar. The hypothesis is that the information on the vertical resolution is hidden within the satellite channels. While previous research reached their limits, we assume that these representations can be learned from a neural network better than by human analysis. We rephrase that paragraph for clarity.

L33 -

– **Comment**: joined -> combined

– **Answer**: Changed to: "A combined use of different instruments to derive comprehensive 3D structures [...]".

L88 -

– **Comment**: The near IR channels also include reflected solar radiation (depending on channel wavelength and time of day)

– **Answer**: We rephrase that sentence to clarify the channel differences.

L109 -

– **Comment**: How is parallax addressed? SEVIRI has an off-nadir view for most points on the disc, while CloudSat always views at nadir. This could cause registration issues between different cloud layers. Changes in resolution might also become an issue closer to the edge of the disc.

– **Answer**: We recognize the importance of the parallax effect for the correction of the geometry, especially when dealing with multiple cloud layers. In this work, we estimate only a small impact on the overall results. First, we assume that neighbouring pixels are more similar to each other than those in far distance. Second, we face the need to downsample the resolution of the CloudSat data to match the MSG SEVIRI resolution. That is why we don't expect the estimated differences of up to 3 pixels to substantially impact the predicted reflectivities. Overall, we estimate the error resulting from geometric inaccuracy to be lower than current model limitations. We will add a remark in the manuscript.

L110 -

– **Comment**: I found the last sentence difficult to follow. What does the 'factor to coarse grain the data' mean here?

– **Answer**: The resolution mismatch between CloudSat and MSG SEVIRI requires an aggregation of CloudSat pixels. We use the maximum reflectivity to reduce the resolution of the radar data. This blurs the sharp contrasts within the radar cross sections (as seen in Figure 4, CloudSat CPR cross sections are blurred compared to the original data) . We rephrase that sentence for clarity.

Fig. 2 -

– **Comment**: Why not low cloud layers? This is explained later on, but could have been included in the description of the radar data (unless I missed it?)

– **Answer**: As stated in lines 123-125, we analyzed the CloudSat quality flag to identify height levels that were affected by noise. This applies especially to the near-ground height levels influenced by the topography and connected radar attenuation. Those height levels were excluded from the model framework. The reduction affects the representation of low clouds

and the cloud base, reducing the model performance in these altitudes. We add an extended explanation in this section in the revised manuscript.

Fig. 3 -

– **Comment**: How is this normalisation done? Is it across the whole image?

– **Answer**: For the joint plot, we computed the value distribution for observed and predicted reflectivities on the test set. The normalisation is done across the whole image using the respective distributions.

L220 -

– **Comment**: Why is the DL model different here? Is this due to a regression-to-the-mean effect for the Res-UNet?

– **Answer**: After revising, we think this was due to an error in the code. The results could differ as a result of the regression-to-the-mean. Since the DL network is fed a lot of nearly cloudfree images, they can lead to a shift of the distribution. In contrast, extreme values have a higher influence on the OLS and RF. This can affect the prediction of positive values. Nevertheless, we change the plot to all models pointing towards the same direction.

L262 -

– **Comment**: I think thin clouds are also difficult to detect from CloudSat, given they might have a small radar reflectivity. Both SEVIRI and CloudSat on their own are able to produce this second, higher peak in cloud top height, suggesting they can detect these thinner clouds.

– **Answer**: For the first sentence, we agree and rephrase that statement. The second part I found to be contradicting. CloudSat in particular can produce this peak whereas our model fails to appropriately reconstruct high reflectivities in higher altitudes (Fig.3). The missing peak might be connected to the imbalance within the data and resulting underestimation of high reflectivities. We rephrase that paragraph for clarity.

Fig. 4 -

– **Comment**: Perhaps blur the Cloudsat data to match the DL resolution? It would also be nice to have 5km marked at the bottom of the y-axis (if that is the case), to highlight this doesn't go to zero.

– **Answer**: Figure 4 shows the blurred CloudSat data after downsampling. The models predict the cross section with a higher blurriness, as a result of the regression-to-the-mean. We will add the lower y-axis label.

L270 -

– **Comment**: So it is (or could be) a simultaneous retrieval, due to the inclusion of the water vapour channels?

– **Answer**: We agree the water vapour channels could lead to this distortion over water bodies. There are only few observations in these regions. Extreme values have a higher influence on the mean. This could be a further source for distortion.

Fig. 7 -

– **Comment**: Much more structure on the Res-UNet results - why is this? Also, the Res-UNet image seems to have almost all the cloud tops at the same altitude, other than a small fraction of low-level cloud in some regions.

– **Answer**: Both are aggregated to a monthly mean, whereas the datasets have a resolution mismatch (0.05 ° for CLAAS-V002E, 0.003 ° for the predictions). The aggregation of the derived CTH contains a higher small-scale variability. The results reflect the lacking model ability to for predict low level and high level clouds (Fig.3). Using a fixed threshold of -15dBZ to identify clouds may not be accurate for all regions of the FD. Combining the threshold with the shifted reflectivity distribution can affect the distribution of the CTH in, smearing out regional differences. As a result of this underestimation of high reflectivities, the mean of the monthly aggregate is shifted towards a lower CTH than for the CMSAF product. CloudSat information is retrieved at the same local time for every region (due to its sun-synchronous orbit) and only applicable to daytime predictions. This can increase the diurnal representation of the reflectivities.

L299 -

– **Comment**: I guess this is much like satellite retrievals in general? Presumably there is actually just a lack of information there?

– **Answer**: Apart from the inaccuracy of the model results itself, we agree this could be traced back to a general lack of information. We rephrase the sentence to clarify.

L300 -

– **Comment**: 5km is quite high from the ground to deal with clutter. It also cuts out a lot of the lower level clouds with a more challenging cloud top height retrieval, which might make the DL method seem better overall? I am not suggesting they should be included (working with just high clouds is an important task), but could be worth mentioning.

– **Answer**: We are aware that low level clouds are important features needed to be represented within the model results. CloudSat suffers from attenuation in low height levels. After analyzing the CloudSat quality flag, we observe that most of the reflectivity values between 0–2.4 km are affected by noise, so we set these values to -25 dBZ. This affects the predictions between 2.4 and 5km height. Due to the regression-to-the-mean and the imbalanced reflectivity distribution, we see a further shift of the distribution towards low reflectivities, especially up to 5 km. Using a modified network architecture and loss function can help to improve the representation of clouds between 2.4–5km, and the calculation of the CTH. We include a statement in the revised manuscript.

L321 -

– **Comment**: I think this reduction in bias should be shown if it is claimed. Those external data sources might decrease bias themselves in some situations (e.g. with a better estimate of water vapour than is available from the SEVIRI IR channels).

– **Answer**: The statement is drawn from the comparison of the CMSAF CTH and the predicted CTH, but we revise the paragraph and add a further quantification.

L325 -

– **Comment**: maybe "remote oceanic regions" instead of "secluded regions above the sea surface"?

– **Answer**: We appreciate that suggestion and change the wording here.

---

## Author Response (AR1)

**Final response**
**(Manuscript "egusphere-2023-1834")**

Sarah Brüning, Stefan Niebler, and Holger Tost

November 15, 2023

**1    Introductory remarks**

In the following sections, we address all comments from the three referees. For every comment, you can find (1) the comment, (2) the author's response (both with lines from initial manuscript), and (3) the author's changes in manuscript. The lines given for **author's changes in the manuscript** refer to the lines in the **revised manuscript**.

Based on the referee comments, we focus the revision on the language. Most of the changes concern the use of technical terms and the description of our method and results. The narrative and main points of the manuscript did not change. We changed the figure labels in the following way:

- 1 -> 1

- 2 -> 4

- 3 -> 3

- 4 -> 5

- 5 -> 6

- 6 -> 8

- 7 -> 9

- New: 2

- New: 10

**2    Referee 1**

**2.1    Comment (general)**

- **Comment**: The most interesting part of the paper to me is the network structure that generates the 3D fields. The U-Net structure is appropriate for the problem at hand, although it is not quite clear from the description how the transformation from 2D to 3D is performed.

- **Author's response**: The transformation process from 2D to 3D is illustrated in Figure 1, Section 2.1.3, and Section 2.2. The matching algorithm stacks the 11 MSG SEVIRI channels of one sample together to one 3D image of 11 x 128 x 128 [C x H x W] pixels. Within this sample, CloudSat crosses a horizontal transect. The location of the cross section is defined as described in lines 104-109. We aggregate the CloudSat pixels on this track by their maximum value to fit the resolution of MSG SEVIRI. As a result, we receive a 3D image of 90 x 128 x 128 [Z x H x W] pixels containing the reflectivities along the cross section. After the downsampling, the transect has a width of one pixel in the 3D image (Fig. 1.3). Pixels outside the transect are assigned as missing values. In that way, we can preserve the location of the CloudSat data during training and use this information to extract the pixels from the 3D predictions to calculate the loss. The loss function is the RMSE

(lines 168-171). As we are restricted to calculate the loss only for predictions along the transect, the loss reflects the model performance on only a small subset of the data (defining it as a sparsely supervised network). Despite these limitations, the final output consists of a full 3D prediction of each tile. Each tile contains a contiguous cloud field and is visually analyzed in Section 3.3 and Figure 5. We add an extended explanation in the revised manuscript.

- **Author's changes in the manuscript**: Section 2.1.3, lines 109ff: "We standardize the data shape by transforming the 2D cross section into a 3D image of 125 x 128 x 128 [Z x H x W] pixels, representing reflectivities along the cross section. After downsampling, the transect becomes one-pixel wide. We label pixels outside the transect as missing values to maintain the CloudSat data's location during training. During evaluation, we use these pixel indices to compute the loss between the CloudSat data and the predicted cross section.".

**2.2   Comment (general)**

- **Comment**: While the network is interesting, I wonder about the usefulness of the model outputs since the model clearly suffers from regression to the mean and blurriness of the results. Also, the uncertainty of the outputs is not estimated. Could you comment a bit on which applications would find the current results useful? And how the above mentioned issues could be improved?

- **Author's response**: As mentioned in lines 242-243 and 305-307, we agree that the network suffers from problems connected to the regression to the mean and the blurriness of results. In the revised manuscript, we will address these problems and possible improvements more straightforward.

  The histogram of the CloudSat data shows a shift towards the background value of -25 dBZ (Fig. 6). To address this imbalance, and, to enhance the model performance, we use the UNet with additional residual connections (Res-UNet) and an automatic learning-rate optimizer. Using a full year of satellite data leads to the generation of approximately 30.000 samples. We reduce the percentage of cloud-free samples face the imbalance between cloudy and clear-sky samples (lines 116-119). Still, a cloudy sample can have a substantial proportion of background values. As reducing the size of the dataset affects the model performance, we randomly flip the samples during training for data augmentation. The goal is to include model invariance to the cloud orientation in the model and avoid overfitting. Although we use techniques to deal with the regression to the mean, the results show room for improvement. All predictions smear out. This is not caused only by the imbalance of the data, but can be led back to the loss function. The RMSE is known to enhance the blurriness of the predictions. An adapted loss may improve the results. Nevertheless, the study suffers from the resolution mismatch between the MSG SEVIRI and the CloudSat data. Aggregating the CloudSat pixels further increases the blurriness of the ground truth as well as those of the predictions (lines 109-112). An improvement can be achieved by using a deeper architecture along satellite data with a higher resolution (e.g., GOES-R, Himawari 8/9, or, in the future, MTG). As stated in lines 318 – 321, the goal of our study is to point out for the first time the possibility to derive continuous 3D clouds using DL. We see the innovation of our work not in the network architecture itself as it was kept simple on purpose. Nevertheless, using powerful architectures like the Vision Transformer as well as an adapted loss can improve the above mentioned issues.

  Despite current performance issues, we see a vast potential for the method in applications of atmospheric and climate sciences. In Section 3.3, we exemplary compute the CTH to evaluate the quality of the predictions for analyzing cloud microphysics (lines 255ff.). Further use-cases comprise e.g., the analysis of cloud organizational patterns, the identification of lightning locations, or a detailed analysis of precipitation onset. We use the CloudSat radar data as ground truth, but the approach can be transferred to different data on a 2D transect, e.g., aerosol measurements. The absence of artifacts between predicted tiles on the FD enables an investigation of 4D movement patterns through space and time (lines 249-254). The data can be used together with simulation results to improve the representation of large-scale cloud dynamics.

- **Author's changes in the manuscript**: We address these issues in Sections 3.2, 3.4, and 4. We discuss further applications and the limitations of the current network in particular in Section 4, lines 337–345, lines 353–366, and lines 382ff.

**2.3 Comment (lines 24-25)**

- **Comment**: "Passive sensors such as geostationary satellites": the statement needs more precision, satellites are not sensors

- **Author's response**: Changed to "While observations from passive sensors on geostationary satellites are limited to monitor the uppermost atmospheric layer from space, [..]".

- **Author's changes in the manuscript**: Section 1, lines 17–18: "Observations from passive sensors on geostationary satellites have a high spatio-temporal coverage, but they are limited to monitor the uppermost atmospheric layer in two-dimensions (2D) (Noh et al., 2022).".

**2.4 Comment (lines 36-37)**

- **Comment**: "The large-scale generability of these methods is expandable since their 3D results are limited to the cloud's spatial vicinity": I don't understand this sentence, please clarify.

- **Author's response**: We agree that this sentence might be unclear. We will omit it as it is not necessary for the central statement of the paragraph. The relevant information can be found in the sentences before and after this one.

- **Author's changes in the manuscript**: Section 1, lines 26–30: "Combining data sources can fill current data gaps (Amato et al., 2020; Steiner et al., 1995). The combined use of different instruments has been investigated before. This research comprises the usage of statistical algorithms (Miller et al., 2014; Seiz and Davies, 2006; Noh et al., 2022), the integration of radiative transfer approaches (Forster et al., 2021; Zhang et al., 2012), or the derivation of the multi-angle geometry of neighboring clouds (Barker et al., 2011; Ham et al., 2015) to reconstruct the cloud vertical column.", and lines 50–51: "To our best knowledge, no extrapolation of 2D radar data to a large-scale 3D perspective was conducted before (Wang et al., 2023; Dubovik et al., 2021).".

**2.5 Comment (line 48)**

- **Comment**: "time efficiency and feasibility": it's not clear to me what this means

- **Author's response**: DL networks do not require feature extraction like traditional ML methods. That reduces the time necessary to start the training and the user generated bias. Also, their performance on big data is better. We change the wording to avoid misunderstandings.

- **Author's changes in the manuscript**: Section 1, lines 36–39: "The recent technological advances enable unprecedented operations (Amato et al., 2020). Deep-Learning (DL) based networks are suitable to identify spatial, spectral, and temporal patterns on big data (Jeppesen et al., 2019; Hilburn et al., 2020). In contrast to traditional ML frameworks, they do not require a manual feature engineering (Le Goff et al., 2017).".

**2.6 Comment (line 80)**

- **Comment**: "orbiting the globe on a sinusoidal track": how does a satellite orbit on a "sinusoidal track"?

- **Author's response**: This refers to the radar track visualization in 2D. Changed to: "The ground truth of the study is derived from an active radar on board the CloudSat satellite which moves on a sun-synchronous orbit."

- **Author's changes in the manuscript**: Section 2.1, lines 70–72: "The ground truth of the study is derived from an active radar on board the CloudSat satellite which moves on a sun-synchronous orbit (CloudSat Data Processing Center, 2023).".

**2.7 Comment (lines 91-92)**

- **Comment**: "resampled to a geographic grid": what kind of grid, a lat-lon one?

- **Author's response**: Yes, data is resampled to a spatial grid in geographic coordinates with a resolution of 0.03 ° (lines 90-92). The missing information will be added.

- **Author's changes in the manuscript**: Section 2.1.1, lines 82–84: "We reformat all satellite images onto a spatial grid with geographic coordinates, employing the global reference system WGS84 (Drönner et al., 2018). Each pixel has a resolution of 0.03 ° in both width (W) and height (H).".

**2.8 Comment (Section 2.1.2)**

- **Comment**: CloudSat is on a sun-synchronous orbit, meaning it sees every location at the same local solar time. This might introduce some diurnal bias to the data; this should be acknowledged.

- **Author's response**: We appreciate that remark and add following: "Since CloudSat follows a sun-synchronous orbit, it receives information on the cloud reflectivity at different locations along the globe always at the same local time. This reduces its ability to reflect the diurnal variability within each region of the AOI.".

- **Author's changes in the manuscript**: Section 2.1.2, lines 97f.: "We note that due to the sun-synchronous orbit of CloudSat, it has a reduced ability to account for diurnal variations within specific regions of the AOI (Stephens et al., 2008)".

**2.9 Comment (line 109)**

- **Comment**: The use of "XY" is confusing, I read this initially as "X times Y" but apparently you mean a diagonal transect through the image? Or did I misunderstand?

- **Author's response**: "XY" refers to the diagonal transect of the CloudSat radar along the pixel dimensions X and Y of the MSG SEVIRI image. We change the naming to "transect" or "cross section" in the text (in figures as [Z,(H,W)]).

- **Author's changes in the manuscript**: Changed in Section 2.1.3, e.g., lines 106–107: "CloudSat flies across a horizontal transect within the satellite scene.", and in Figure 1 (c).

**2.10 Comment (line 125)**

- **Comment**: "smoothing" should probably be "filtering"

- **Author's response**: We agree and change to "filtering".

- **Author's changes in the manuscript**: We changed the sentence in Section 2.1.4, lines 123–125: "We use the CloudSat quality index to identify noisy pixels. Pixels with a quality index lower than six were set to a background value of -25 dBZ to reduce noise (Marchand et al., 2008).".

**2.11 Comment (line 137)**

- **Comment**: The use of the word "delineate" here and a couple of other places in the places seems incorrect

- **Author's response**: Changed "delineate" to "predict" (line 7, line 137) or "estimate" (line 81).

- **Author's changes in the manuscript**: Changed the wording, e.g., in Section 2.2, lines 134–135: "Former studies using the Res-UNet dealt with the classification of tree species (Cao and Zhang, 2020) or the prediction of precipitation (Zhang et al., 2023).".

**2.12   Comment (line 159)**

- **Comment**: I would like some more details on how the network structure maps the 2D input fields to the 3D output fields. Is the channels dimension used transformed to the Z dimension of the output?

- **Author's response**: See Figure 1 and Section 2.2. The MSG SEVIRI data consists of a 2D field for each satellite channel. During the matching scheme, these channels are stacked to one 3D image of 11 x 128 x 128 pixels. On the encoder side of the network, the channel dimension of the MSG SEVIRI input is expanded to the proposed filter size of 256. On the decoder side, these filters are reduced and mapped to a model output size of 90 x 128 x 128 pixels. As a result, the channel dimension of the output equals the Z dimension of the CloudSat ground truth. We will add an extended explanation in the revised manuscript.

- **Author's changes in the manuscript**: Section 2.1.3, lines 105-113: "We prepare the matched image-profile pairs for further processing. To do this, we combine the 11 MSG SEVIRI channels into a single 3D image with dimensions 11 x 128 x 128 [C x H x W] pixels. CloudSat flies across a horizontal transect within the satellite scene. It has a higher native resolution than MSG SEVIRI. To align the datasets, we downsample the radar pixels by aggregating them based on the local maximum reflectivity. This adjusts the CloudSat pixels to the MSG SEVIRI resolution of 0.03 ° but leads to some loss of sharp contrast in radar pixels (Jordahl et al., 2020). We standardize the data shape by transforming the 2D cross section into a 3D image of 125 x 128 x 128 [Z x H x W] pixels, representing reflectivities along the cross section. After downsampling, the transect becomes one-pixel wide. We label pixels outside the transect as missing values to maintain the CloudSat data's location during training. During evaluation, we use these pixel indices to compute the loss between the CloudSat data and the predicted cross section.". Section 2.2, lines 152–153: "The final 1 x 1 convolution maps the output to 90 x 128 x 128 pixels, representing the 90 height levels of the radar cross-section (Jeppesen et al., 2019).".

**2.13   Comment (lines 163-164)**

- **Comment**: How is the 3D scene predicted by the network compared to the CloudSat data during training? CloudSat only gets a 2D vertical cross section of the scene. Is only part of the scene selected for comparison? Also, what loss function do you use for training?

- **Author's response**: Seems to be similar to the question dealing with the transformation from 2D to 3D in the upper part of the text. In summary, we transform the 2D cross section of CloudSat to a 3D image of 90 x 128 x 128 pixels during the matching scheme. This 3D image has the same size as the model output after training. For the CloudSat 3D image, only pixels along the transect hold finite values. These can be used to filter both images and calculate the RMSE between the observed and predicted cross section (sparse supervised network). We will add this information in the revised manuscript.

- **Author's changes in the manuscript**: Section 2.3.1, lines 166–171: "The model performance is evaluated during the training process by calculating the root-mean-square error (RMSE). The RMSE is equally able to penalize misses and false alarms (Lee et al., 2021). As described in Section 2.1.3, we preserve the pixel indices of the CloudSat cross section within each image-profile pair during training. We use the location of these pixels to filter the observed and predicted transect. The loss is calculated along the filtered cross sections. Since it is only evaluated on a small subset of 10 % of all pixels, we have a sparse regression task (Wang et al., 2020). We cannot quantify the model performance on the full 3D prediction of the cloud field.".

**2.14   Comment (line 176)**

- **Comment**: "Both models": unclear which models this refers to

- **Author's response**: Refers to the pixel-based methods mentioned in line 172. We rewrite the sentence for clarity.

- **Author's changes in the manuscript**: Section 2.3.1, lines 173–176: "The results of the Res-UNet are compared against two competitive methods (Drönner et al., 2018). First, we predict the radar reflectivity by an ordinary least squares model with multiple regression output (OLS). The 11 satellite channels were used as independent predictor variables. The output is a 1D cloud column. Second, a Random-Forest (RF) regression is applied (Breiman, 2001).".

**2.15 Comment (line 181)**

- **Comment**: "pictures" is used incorrectly here.

- **Author's response**: Changed to "The RF is a supervised ML algorithm which provides robust results when working with environmental datasets in the natural sciences [. . . ].".

- **Author's changes in the manuscript**: Section 2.3.1, lines 176–177: "The RF is a supervised ML algorithm suitable when working with environmental datasets in the natural sciences (Boulesteix et al., 2012).".

**2.16 Comment (lines 189-190)**

- **Comment**: In the joined 2400 x 2400 pixel 3D prediction, is the field continuous at the borders of the 100 x 100 pixel tiles? Or do you see discontinuities or artifacts?

- **Author's response**: The joined 3D prediction consists of a continuous field without artifacts. As shown in Figure 5, each of the sub-figures (b) – (d) represents a combination of several 100 x 100 pixel tiles ( 2.5 ° on the Lat-Lon grid). The results are described in Section 3.3, lines 250 – 252 and will be explained in more detail in the revised manuscript.

- **Author's changes in the manuscript**: Section 2.3.2, lines 189–191: "For the FD prediction, we divide the FD into overlapping subsets of 128 x 128 pixels. These subsets are processed and fed into the network. The output is a 3D reflectivity image of 90 x 100 x 100 pixels [Z x H x W], which equals 2.5 ° on the MSG SEVIRI grid. We merge the tiles to cover the whole satellite AOI. Between the tiles, there is no overlap.". The results are discussed in Section 3.3.

**2.17 Comment (line 220)**

- **Comment**: "That said" seems out of place here - please revise.

- **Author's response**: Changed to: "[. . . ] The DL network indicates an underestimation of high reflectivities and an overestimation of low reflectivities for low-level clouds.".

- **Author's changes in the manuscript**: Section 3.1, lines 222-223: "The results indicate an overestimation of high reflectivities and an underestimation of low reflectivities, especially for low-level clouds.".

**2.18 Comment (line 231)**

- **Comment**: I don't understand that "denominational structure" means here.

- **Author's response**: Refers to the results for the OLS and RF in Figure 4. In contrast to the DL results, the reconstructed cross section is not as smooth but shows a rather fragmented structure.

- **Author's changes in the manuscript**: Section 3.2, lines 257-259: "While the 2D profiles of the Res-UNet are smooth, the RF and OLS lead to a fragmented structure with a high value variability between the single pixels of the transect (I, IV). The examples show an inaccurate reconstruction of shallow clouds and multi-layer clouds for the OLS and RF.".

**2.19 Comment (lines 262-263)**

- **Comment**: High, thin ice clouds may also not be observed by CloudSat due to being under the minimum detectable reflectivity.

- **Author's response**: Thank you that remark. We will change the text in the revised version.

- **Author's changes in the manuscript**: We omit that statement in the revised version and wrote instead (Section 3.2, line 234): "Above 15 km, we have few CloudSat observations > 15 dBZ (Fig. 2, a).".

**2.20 Comment (Figure 7)**

- **Comment**: Maybe you could add a panel showing the difference of a and b to illustrate the biases better.

- **Author's response**: Interesting suggestion, we add that figure in the revised manuscript.

- **Author's changes in the manuscript**: Figure 7 is now Figure 9. Figure 10 (page 21) shows the difference between the observed (a) and predicted (b) CTH from Figure 9.

**2.21 Comment (line 288)**

- **Comment**: "Leaving out the affected channels downgrades the overall performance": it would be good to see something to demonstrate this.

- **Author's response**: We tried a model setup without VIS. Due to its poor results, we did not include it in the manuscript for brevity. In the revised manuscript, we will rewrite the text in this paragraph.

- **Author's changes in the manuscript**: Section 4, lines 360–366: "In this study, we only derive daytime predictions. This is due to the influence of solar radiation in visible spectrum channels (VIS) (Hilburn et al., 2020; Jeppesen et al., 2019). Additional distortions may arise from VIS channels, as satellite data only represent the uppermost cloud layer. Depending on the location, they can be highly influenced by the surface albedo (Drönner et al., 2018). Training a model without the VIS channels can help to achieve predictions independent of the daytime. Reducing the extent of the AOI can mitigate the geographic performance differences but limits the applicability of the network. Training regional models and adjusting the loss function and model architectures offer potential solutions to improve the results of the 3D cloud tomography.".

**2.22 Comment (lines 292-293)**

- **Comment**: "In contrast to pixel-based DL methods like the CNN or CGAN, the Res-UNet utilizes a larger receptive field preserving the spatial dimensionality and global context information during the training routine." This is not a correct statement regarding the CNN or CGAN architectures. CNN architectures can also achieve large receptive fields and global context using downsampling. In fact the UNet itself is a type of CNN - its distinguishing feature is the addition of skip connections to preserve resolution. As for the CGAN, it refers to a certain training setup of generative models that could be implemented with either normal CNNs or with (Res-)UNets.

- **Author's response**: We agree and rewrite the sentence: "Like other CNN architectures, the Res-UNet preserves the spatial dimensionality and global context information during the training routine (Wang et al., 2022). Compared to pixel-based methods like the OLS, it reconstructs the spatial connectivity between the pixels more accurately and enhances the representation of the continuous cloud field. [...]".

- **Author's changes in the manuscript**: Section 4, lines 330–332: "The Res-UNet makes predictions based on 3D images, preserving the spatial details and global context during training (Wang et al., 2022). The error of the model varies depending on cloud structure within the radar cross-section. Compared to pixel-based approaches like OLS, the Res-UNet better reconstructs the pixel connectivity.".

**2.23 Comment (lines 312-313)**

- **Comment**: Approximately 1 km resolution is also already available from the GOES-R series and Himawari 8/9 satellites.

- **Author's response**: We know about the resolution of these satellites, but we decided to stick with the MSG satellite to fit our main study area. For a potential global composite of the 3D predictions, data from GOES-R and Himawari 8/9 may help to improve the results. We add this information and change lines 311-314: "At the moment, a compromise on the resolution is necessary to obtain predictions centered over Europe and Africa. However, newly emerging instruments offer an enticing prospect to tackle this information loss. While comparable data sources like the GOES-R series or the Himawari 8/9 satellites already provide data in

a resolution of 1 km, the recently launched satellite Meteosat Third Generation by EUMETSAT (Holmlund et al., 2021) is expected to close the data gap. Together, they can be leveraged to investigate a 4D reflectivity field through space and time".

- **Author's changes in the manuscript**: Section 4, lines 381–387: "Currently, a compromise on the resolution is necessary to obtain predictions for Europe and Africa. However, promising new instruments are emerging. While data from comparable sources like the GOES-R series and the Himawari 8/9 satellites already offer a 1 km resolution, the recently launched Meteosat Third Generation satellite by EUMETSAT allows to close the gap and enables a more precise representation of individual clouds (Holmlund et al., 2021). Although our approach currently focuses on a region centered around 0 ° longitude, we can apply the same framework to other geostationary satellites, potentially achieving global 3D cloud coverage throughout the troposphere. The predicted cloud field can be valuable for time series analysis, enabling the tracking of clouds in four dimensions across space and time.".

**2.24 Comment (lines 321-322)**

- **Comment**: "Since it is independent of external or interconnected data sources, the bias within the data is reduced.": unclear sentence, I'm not sure how the latter follows from the former.

- **Author's response**: The operational CM SAF CTH is computed using the MSG SEVIRI satellite data as well as derived products and additional data. Each of them may bring their own bias, potentially multiplying their effects on the final CTH. In contrast, our CTH is based only on the predicted reflectivity. In that way, we can minimize the influence of additional data sources. We will rewrite this explanation in the revised manuscript.

- **Author's changes in the manuscript**: Section 3.4, lines 325–329: "Even though the comparison of the CTH points out regional differences, the predictions can be used to represent the CTH pattern on the FD. The CLAAS-V002E1 data is computed using the MSG SEVIRI satellite channels as well as derived products and additional data. Each of them bring their own bias, potentially multiplying their effects on the final CTH. In contrast, our CTH is based only on the predicted reflectivity. In that way, we can minimize the influence of additional data sources.", and Section 5, lines 396–397: "Using only the predicted reflectivity, we derive the CTH without external data sources.".

**3 Referee 2**

**3.1 Comment (general)**

- **Comment**: In my opinion, any application of machine learning to scientific research questions requires some level of uncertainty quantification. I appreciate that this might go beyond the scope of the paper, but estimating model uncertainties would greatly improve the trustworthiness of the results, and likely help inform how (and for which scenarios) the model needs improving.

- **Author's response**: We agree this is an interesting idea. We did not include a quantification of model uncertainties since it seemed out of scope of our initial submission. We will evaluate whether and how to integrate a form of uncertainty quantification in the revised manuscript.

- **Author's changes in the manuscript**: We did not include a quantitative uncertainty estimation which is out of scope for this paper. Instead, we discuss current limitations of the network and possible solutions in Section 4.

**3.2 Comment (general)**

- **Comment**: The writing is at times too verbose and unclear. I am including a number of specific comments below, but I would suggest going through the paper again and streamlining the narrative. At times, the section headings could be more specific. Some of them (e.g. "Comprehensive predictions") are not particularly descriptive of the work done.

- **Author's response**: We revise the manuscript with particular attention to this issue.

- **Author's changes in the manuscript**: Throughout the text, we edited the language to be more clear and concise. We modified the headings of the subsections in sections 2 and 3.

**3.3 Comment (general)**

- **Comment**: Please add tables of the training and hyperparameters in the paper (or better Supplementary Information). Most importantly, please include information about the number of parameters of each of your models.

- **Author's response**: The tables will be added in the revised manuscript.

- **Author's changes in the manuscript**: The tables are added in the appendix A and B (page 25) as Table A1 (Network parameters) and Table B1 (Summary of model parameters).

**3.4 Comment (line 9)**

- **Comment**: Abstract: "average error" is a bit vague and not the same as RMSE. Please revise. Furthermore, it is not clear what "total value range" in this sentence is referring to.

- **Author's response**: Changed to: "The RMSE after training is 3.41 dBZ which equals a difference of 7.5 % compared to a reflectivity scale between -25 and 20 dBZ."

- **Author's changes in the manuscript**: Abstract, lines 6–7: "Our RMSE accounts for 2.99 dBZ. This corresponds to 6.6 % error on a reflectivity scale between -25 and 20 dBZ.".

**3.5 Comment (line 27)**

- **Comment**: "[...] spatially and temporally limited perspective [...]." Please expand on what the spatial and temporal characteristics of the different instruments used in this work are. Specifically, I think the authors could expand on the benefit of geostationary satellites when it comes to temporal resolution, as polar-orbiting satellites often require 1+ days to observe the same area again.

- **Author's response**: The spatial and temporal characteristics of the specific instruments are described in Section 2.1.1 and 2.1.2. We agree that some points may be missing. In the revised manuscript, we add a more detailed comparison.

- **Author's changes in the manuscript**: Section 1, lines 22–25: "The radar receives detailed information on the cloud column along a 2D cross section with a high ground resolution and constant sun illumination. Due to its sun-synchronous orbit, it observes the same spot at the same local time. Compared to geostationary satellites, the active radar does not provide a continuous spatial and temporal coverage (Wang et al., 2023).", and Section 2.1.2, lines 97f.: "We note that due to the sun-synchronous orbit of CloudSat, it has a reduced ability to account for diurnal variations within specific regions of the AOI (Stephens et al., 2008).".

**3.6 Comment (line 29)**

- **Comment**: "While this analysis often rests upon subjective labeling or fixed thresholds [...]." Please add example citations to previous work.

- **Author's response**: We change the text adding examples for the retrieval of the cloud optical depth (https://doi.org/10.1175/2011JA and the cloud effective radius (https://doi.org/10.5194/acp-20-1131-2020).

- **Author's changes in the manuscript**: Section 1, lines 19–21: "By using the satellite's specificity at different wavelengths (Thies and Bendix, 2011), and subjective labeling or fixed thresholds (Platnick et al., 2017), we can estimate cloud physical properties like the cloud optical thickness (Henken et al., 2011) or the effective radius (Chen et al., 2020).".

**3.7 Comment (line 33)**

- **Comment**: "joined" ==> joint

- **Author's response**: Changed to: "A combined use of different instruments to derive comprehensive 3D structures [...]".

- **Author's changes in the manuscript**: Section 1, lines 26–30: "Combining data sources can fill current data gaps (Amato et al., 2020; Steiner et al., 1995). The combined use of different instruments has been investigated before. This research comprises the usage of statistical algorithms (Miller et al., 2014; Seiz and Davies, 2006; Noh et al., 2022), the integration of radiative transfer approaches (Forster et al., 2021; Zhang et al., 2012), or the derivation of the multi-angle geometry of neighboring clouds (Barker et al., 2011; Ham et al., 2015) to reconstruct the cloud vertical column.".

**3.8 Comment (line 36)**

- **Comment**: I am not sure what "generability" means.

- **Author's response**: This sentence is unclear, we omit it. The sentence before and after (lines 35, 37) are sufficient for our statement (that, to our best knowledge, no large-scale 3D interpolation of the cloud reflectivity from radar exists).

- **Author's changes in the manuscript**: Omitted in paragraph at lines 25–30. Instead, we add in Section 1, lines 50–51: "To our best knowledge, no extrapolation of 2D radar data to a large-scale 3D perspective was conducted before (Wang et al., 2023; Dubovik et al., 2021).".

**3.9 Comment (line 54)**

- **Comment**: I would suggest including other works, such as https://arxiv.org/abs/1911.04227, when discussing cloud classification.

- **Author's response**: We appreciate that remark and revise the citations.

- **Author's changes in the manuscript**: Section 1, lines 41–45: "So far, cloud properties have been investigated by DL algorithms in various applications. These comprise the detection (Drönner et al., 2018) and segmentation of cloud fields (Jeppesen et al., 2019; Lee et al., 2021; Le Goff et al., 2017; Tarrio et al., 2020; Cintineo et al., 2020), or the classification of distinct cloud types from meteorological satellites and aerial imagery (Marais et al., 2020; Wang et al., 2023). Zantedeschi et al. (2022) used a neural network to bring together information from an active radar and high resoluted satellite data to reconstruct cloud labels.".

**3.10 Comment (lines 70+)**

- **Comment**: The sentence is very long and hard to follow. Please rephrase.

- **Author's response**: Changed to: "Former studies had a focus on reconstructing the 1D or 2D cloud column. In contrast, we apply a DL framework to predict the radar reflectivity not only along the radar cross section, but on the satellite full disk (FD)."

- **Author's changes in the manuscript**: Section 1, lines 59–60: "Previous studies focused on reconstructing the 1D cloud column or 2D cross section. In contrast, our approach utilizes a DL framework to predict the radar reflectivity not only along the radar cross section but across the entire satellite full disk (FD).".

**3.11 Comment (line 78)**

- **Comment**: "[...] originates a geostationary satellite." ==> "from" missing.

- **Author's response**: Changed to: "The input data for the network originates from a geostationary satellite [...]".

- **Author's changes in the manuscript**: Section 2.1, lines 66–67: "The input data for the neural network originates from a geostationary satellite.".

**3.12 Comment (line 84)**

- **Comment**: The EUMETSAT abbreviation was already used in line 78, before it was defined here.

- **Author's response**: Moved the definition to line 78.

- **Author's changes in the manuscript**: Section 2.1, lines 68–70: "We use data from the European Organisation for the Exploitation of Meteorological Satellites (EUMETSAT) Spinning Enhanced Visible and InfraRed Imager (SEVIRI) instrument on the Meteosat Second Generation (MSG) satellite (EUMETSAT Data Services, 2023).".

**3.13 Comment (Figure 1)**

- **Comment**: It would be great if you could add information about the size of the boxes shown in Part 1, to make it more clear how the matching algorithm works. Furthermore, it was confusing to see 90 height levels in the figure, when the text previously described CloudSat to have 125 height levels. Could you also comment on the missing data shown in the CloudSat profile?

- **Author's response**: We revise the figure adding an extended description of the matching algorithm. The boxes have a size of 128 x 128 pixels. The missing data in the profile represents every pixel with a value of -25 dBZ (lines 125-127). In Figure 1, these values are transparent to highlight the profile itself. We add this explanation in the figure caption. The reduction of the height levels is described in lines 123-124, but we restructure the section for clarity.

- **Author's changes in the manuscript**: The box sizes and the transparency description are added in Figure 1 and its caption. The height level reduction is described in Section 2.1.2, lines 92–97: "We use the reflectivity transects as the ground truth to evaluate the model results. In the subsequent steps, we adjust the radar's height levels. The lower altitudes, specifically those between 0 and 3 km, are influenced by the topography and a radar signal weakening due to attenuation (Marchand et al., 2008). To enhance the model's performance, we omit the lowest 10 height levels. Since we notice a significant imbalance between clear-sky and cloudy pixels, we exclude the predominantly cloud-free areas within the upper 25 height levels (Stephens et al., 2008). The Z-dimension now encompasses 90 height levels ranging from 2.4 km to 24 km.".

**3.14 Comment (line 114)**

- **Comment**: "Extracted satellite samples display the physical predictors fed into the network [...]." Please rephrase, as it is not clear to me what you mean.

- **Author's response**: Changed to: "The matching algorithm extracts the image-profile pairs. We use the information of the satellite channels within these pairs to reconstruct the vertical cloud distribution."

- **Author's changes in the manuscript**: Section 2.1.3, line 100: "We obtain training data for our study by aligning MSG SEVIRI scenes with CloudSat radar data as shown in Figure 1 (1).". The input data is defined in Section 2.1, lines 66–67: "The input data for the neural network originates from a geostationary satellite.".

**3.15 Comment (line 115)**

- **Comment**: I would rename "samples" as "matched image-profile pairs" or similar to be more exact.

- **Author's response**: We appreciate that suggestion and revise the phrase.

- **Author's changes in the manuscript**: Renamed to "matched image-profile pairs" throughout the text (e.g., line 105).

**3.16   Comment (line 117)**

- **Comment**: How come you have no test set? Validation sets are great, but susceptible to "human gradient descent", since we usually justify model modifications by improved performances on the validation set. This doesn't necessarily mean that the model is better, it just means that performance on these specific examples is improved.

- **Author's response**: The model is trained and validated on data from 2017 (lines 115-116). We use nine months for training (here January-September) and three months for validation (October-December). Due to limited resources, data from May 2016 (n=1500) is used as a test set to evaluate the model performance and to calculate the CTH. We will revise the text and the reported figures to ensure they represent the performance on the test set. In this study, we apply a standard architecture with slightly modified learning-rate and weight decay. In the manuscript, we add a more detailed training protocol.

- **Author's changes in the manuscript**: Description of the test set in Section 2.1.4, lines 117–118: "Our test set is derived from data in May 2016, from which the matching algorithm extracts 1500 image-profile pairs.". It is used to evaluate the model, e.g., Section 3.1, lines 206–207: "We analyze the ability of the three models (Res-UNet, OLS, and RF) to reconstruct the cloud vertical distribution for the test dataset in May, 2016 (Sect. 2.3.1).". We checked all figures and the results in Section 3 to refer to the test set.

**3.17   Comment (line 123)**

- **Comment**: How was the reduction from 125 to 90 height levels done?

- **Author's response**: We analyzed the CloudSat quality flag for the radar profiles to identify height levels with a high proportion of noise (lines 124-127). The original radar scene contains a high amount of noisy pixels up to 2-3km height. That is why we crop off the lower 10 height levels. The reduction affects the representation of low clouds and the cloud base, reducing the model performance in these altitudes. We find an additional cloud free region > 20 – 30 km (https://doi.org/10.1029/2008JD009982, line 19). To reduce the proportion of cloud-free pixels, we cropped these upper height levels off. The final Z-dimension consists of 90 height levels (10 - 100, between 2.4 km and 24 km). We add an extended explanation in the revised manuscript.

- **Author's changes in the manuscript**: The explanation can be found in Section 2.1.2, lines 92–97 (see Comment 3.13): "We use the reflectivity transects as the ground truth to evaluate the model results. In the subsequent steps, we adjust the radar's height levels. The lower altitudes, specifically those between 0 and 3 km, are influenced by the topography and a radar signal weakening due to attenuation (Marchand et al., 2008). To enhance the model's performance, we omit the lowest 10 height levels. Since we notice a significant imbalance between clear-sky and cloudy pixels, we exclude the predominantly cloud-free areas within the upper 25 height levels (Stephens et al., 2008). The Z-dimension now encompasses 90 height levels ranging from 2.4 km to 24 km.".

**3.18   Comment (line 140)**

- **Comment**: "By seeking non-linear approximations of between the input and the output data [...]." Please rephrase.

- **Author's response**: Move the sentence up to line 134 and changed to: "Neural networks have the potential to capture highly complex relationships between input and output data. The Res-UNet displays a modified framework designed for the use-case of remote sensing data. [...]"

- **Author's changes in the manuscript**: Section 2.2, line 131: "Neural networks can capture highly complex relationships between input and output data (Lee et al., 2021).".

**3.19   Comment (line 163)**

- **Comment**: "As flipped images are perceived as new samples, we enhance the amount of training data by giving all samples a chance of 25 % to be either vertically or horizontally rotated." Please comment on whether these

transformations are valid for the context of satellite measurements, especially CloudSat, with its ascending and descending orbits. Are the two acquisitions totally equal, in that flipping one can simulate the other?

- **Author's response**: The data augmentation used here is not meant to simulate ascending/descending orbits. Instead, by applying random flipping, we aim at including invariance to the cloud orientation in the model and avoid overfitting. A structured analysis of differences between ascending/descending CloudSat orbits is out of scope for this paper. Since data from both orbits are included in the training set we don't expect a different performance of the model on either orientation.

- **Author's changes in the manuscript**: Section 2.2, lines 162-164: "To enhance the amount of training data, we give all input data a chance of 25 % to be rotated by 90 ° (Jeppesen et al., 2019). These flipped images are perceived as new samples. The goal is to increase the model invariance to the orientation of the radar cross section.".

**3.20   Comment (Equation 3)**

- **Comment**: Either define all variables in the equation, or remove from the paper, as the RMSE is a relatively standard quantity.

- **Author's response**: We remove the equation in the revised manuscript.

- **Author's changes in the manuscript**: Removed the equation from manuscript (Section 2.3.1).

**3.21   Comment (lines 173-174)**

- **Comment**: Rephrase "horizontal diagonal".

- **Author's response**: Changed to "radar transect" or "radar cross section".

- **Author's changes in the manuscript**: Renamed to "radar transect" or "cross section" throughout the text (e.g., line 169).

**3.22   Comment (Figure 3)**

- **Comment**: Please explain how many samples the joint plot was calculated over.

- **Author's response**: The plot was calculated over all samples of the test set (n=1500). We add the number of samples in the revised figure.

- **Author's changes in the manuscript**: Section 3.1, lines 217–219: "We analyze the difference between the observed and predicted reflectivities by a two-dimensional joint distribution plot. For this purpose, we calculate the density distribution of the reflectivity between 2.4–24 km. Here, we use a bin size of 1 dbZ and 240 m height, respectively (Steiner et al., 1995). All distributions are calculated on the test dataset and normalized by the distribution size (n = 1500).".

**3.23   Comment (lines 218+)**

- **Comment**: This sentence is very hard to follow. Please re-write.

- **Answer**: We revise this paragraph. This sentence will be changed to: "In the joint plot, values around 0 represent a high agreement between the observed and predicted distribution. On these height levels, the observed reflectivities are most accurately reconstructed. We observe areas of high agreement in shape of a curved line reaching from high to low altitudes for all of the three models."

- **Author's changes in the manuscript**: We rephrased that paragraph in Section 3.1, lines 220-222: "Predictions differ from the original radar data, especially for values > 0 dBZ and in low altitudes (Fig. 3). The results indicate an overestimation of high reflectivities and an underestimation of low reflectivities, especially for low-level clouds.".

**3.24 Comment (lines 220+)**

- **Comment**: Do you have any idea why you observe different performance trends for your pixel-based and Res-UNet approaches?

- **Author's response**: After revising, we think this was due to an error in the code. The results could differ as a result of the regression-to-the-mean. Since the DL network is fed a lot of nearly cloudfree images, they can lead to a shift of the distribution. In contrast, extreme values have a higher influence on the OLS and RF. This can affect the prediction of positive values. Nevertheless, we change the plot to all models pointing towards the same direction.

- **Author's changes in the manuscript**: In the revised version of Figure 3, the directions are the same for the Res-UNet, the OLS, and the RF (page 12).

**3.25 Comment (line 226)**

- **Comment**: How were the four samples chosen?

- **Author's response**: Those four samples were randomly chosen from the test set. We add the missing information in the revised manuscript. We add this information in the manuscript.

- **Author's changes in the manuscript**: Figure 4 is now Figure 5 (page 15). Section 3.2, line 240: "Figure 5 shows the predicted and observed reflectivity along the radar transect for four randomly chosen samples."

**3.26 Comment (line 228)**

- **Comment**: What do you mean by "transferability" in this context?

- **Author's response**: Refers to the ability of the network to better represent the horizontal and vertical position of the clouds along the transect. Wee see this is unclear and omit the sentence, as the information is more precisely expressed in the sentences before and after.

- **Author's changes in the manuscript**: Section 3.2, lines 240–241: "For all models, the reconstructed cloud signal is predicted at the right horizontal location along the cross section." and lines 255–257: "The OLS (RF) fails to accurately reconstruct the vertical extent in all transects. Instead, the reflectivity is uniform along the cloud column. We see a continuous cloud signal between 5-–15 km (Fig. 5, c). Contrasting, the Res-UNet predicts the vertical variability more precisely (Fig. 5, b)".

**3.27 Comment (line 231)**

- **Comment**: "A denominational structure [...]." I am not sure what you mean by this.

- **Author's response**: Refers to the results for the OLS and RF in Figure 4. In contrast to the DL results, the reconstructed cross section is not as smooth but shows a rather fragmented structure.

- **Author's changes in the manuscript**: Section 3.2, line 256–259: "While the 2D profiles of the Res-UNet are smooth, the RF and OLS lead to a fragmented structure with a high value variability between the single pixels of the transect (I, IV). The examples show an inaccurate reconstruction of shallow clouds and multi-layer clouds for the OLS and RF.".

**3.28 Comment (Figure 4)**

- **Comment**: Please label each subplot, and refer to the subplots as you discuss the results in the main text to make it easier to follow your arguments.

- **Author's response**: Thank you for the remark, we add this in the revised manuscript.

- **Author's changes in the manuscript**: Figure 4 is now Figure 5 (page 15). When discussing the results, the model subplots are described as (a)–(d), the profiles as (I)–(IV).

**3.29  Comment (line 233)**

- **Comment**: "[...] the Res-UNet shows more robust results [...]." I am not 100 % convinced that the RMSE and visual inspection of the results qualify this sentence. Have you looked at any other metrics, or quantitatively studied performance as a function of cloud type for more than a couple of samples?

- **Author's response**: We did not derive the cloud type as it was not within the scope of the study. But we used not only a few samples for the evaluation but we calculated the RMSE for all models on the test set (n=1500). Over all height levels, the RMSE for the Res-UNet decreases by  30-35 % compared to the OLS & RF. This percentage is used to support our hypothesis.

- **Author's changes in the manuscript**: We base our conclusion on the performance of the Res-UNet as described in Section 3.2, lines 229–231: "The results point out an overall lower RMSE for the Res-UNet than for the OLS and RF (Fig. 4). The mean RMSE varies between 2.99 dBZ for the Res-UNet, 4.1 dBZ (RF), or 4.58 dBZ (OLS). On a dBZ scale between -25–20 dBZ, this is equivalent to an error of 10.1 % (OLS), 9.1 % (RF), or 6.6 % (Res-UNet).", and lines 238–239: "Over all height levels, the Res-UNet has the lowest RMSE of the three models. Compared to the OLS (RF), the mean RMSE of the Res-UNet is reduced by 34,8 % (27,1 %).".

**3.30  Comment (lines 235+)**

- **Comment**: I am not sure I follow your discussion about quality flags. Please clarify.

- **Author's response**: We refer to the internal CloudSat flag that describes the quality of the received reflectivity (lines 125-127). Reflectivity values are filtered by a minimum threshold of six and set to -25 dBz (our background value). As this affects almost all values in low altitudes, our network reconstructs only the cloud signal above 5 km height. We will revise this paragraph.

- **Author's changes in the manuscript**: Described in Section 2.1.4, lines 123–125: "We use the CloudSat quality index to identify noisy pixels. Pixels with a quality index lower than six were set to a background value of -25 dBZ to reduce noise (Marchand et al., 2008).", and again in Section 3.2, line 231–233: "Between 2.4-–5 km, the RMSE is 0. This is due to the lack of CloudSat observations after filtering noisy pixels (Fig. 2, a). Between 5—7 km, the RMSE increases to up to 8 dBZ for the Res-UNet, 10 dBZ for the RF, and 12 dBZ for the OLS (Fig. 4).".

**3.31  Comment (lines 238-239)**

- **Comment**: "[...] this leads to a lower model uncertainty." Do you mean "lower error"? Model uncertainties to me means quantification of how certain a model is in its predictions.

- **Author's response**: Yes, this refers to the model error and the difference between the observed and predicted distribution. We change the manuscript accordingly.

- **Author's changes in the manuscript**: Section 3.2, lines 233–238: "In higher altitudes, the performance of all models improves. The RMSE decreases to 4 dBZ (5.7 dBZ, 6 dBZ) for the Res-UNet (RF, OLS) at 15 km and reaches its minimum at 22 km (24 km for OLS and RF). Above 15 km, we have few CloudSat observations > 15 dBZ (Fig. 2, a). We observe a lower model error (Fig. 4) and reduced difference between the distributions (Fig. 3) in these height levels for all three models. The improved performance can be led back to the superior number of background reflectivities or the presence of more uniform clouds, like extended tropical cirrus.".

**3.32  Comment (line 239)**

- **Comment**: "That said [...]." This sentence isn't quite clear to me. Please rephrase/clarify.

- **Author's response**: Changed to: "Leaving out noisy pixels is needed to improve the model performance. At the same, this results in a loss of information in low altitudes. This issue is reflected within the results. [...]"

- **Author's changes in the manuscript**: Section 3.2, line 237: "The improved performance can be led back to the superior number of background reflectivities [...]", and Section 4, lines 370–373: "High reflectivities tend to be underestimated due to noise near the ground (Stephens et al., 2008). To mitigate this, we exclude affected height levels, but this results in incomplete model predictions between 0—5 km (Fig. 2). Reducing noise is crucial for improving the performance of DL applications in remote sensing (Enitan and Ilesanmi, 2021).".

**3.33 Comment (line 260)**

- **Comment**: "The first peak [...]." It would be really helpful if you could refer to the labels of the subplots when discussing the results.

- **Author's response**: We see this is confusing and revise the section to be more clear.

- **Author's changes in the manuscript**: Section 3.4 (page 20), Figure 8 (formerly Figure 6). The subplot labels (a)–(d) are used when discussing the results.

**3.34 Comment (lines 264+)**

- **Comment**: "These channels are identified as essential information [...]." It would be great if you could show evidence for this, or expand the discussion on what makes you draw this conclusion.

- **Author's response**: This statement is based on a different model setup without VIS. In the initial submission, we did not add this for brevity, but we revise the paragraph and add supporting information.

- **Author's changes in the manuscript**: Section 3.4, lines 301–312 describes the differences between observed and predicted reflectivities. In the revised manuscript, we omit the sentence above.

**3.35 Comment (line 267+)**

- **Comment**: "Comparing [...] reveals an overall high agreement." Do you have any quantitative comparisons between your model outputs and the CTH from CLAAS, or is this mainly from visual inspection?

- **Author's response**: The initial comparison was visual. In the revised manuscript, we add a quantitative analysis.

- **Author's changes in the manuscript**: Section 3.4, Figure 10 (page 21) shows the differences between the observed and predicted CTH, and the geographic distribution of the error. The Figure is described in lines 313ff.

**3.36 Comment (line 277)**

- **Comment**: "The approach offers [...]." Which approach are you referring to?

- **Author's response**: Refers to computing the CTH only by predicted reflectivities. We change that paragraph to be more precise.

- **Author's changes in the manuscript**: Section 3.4, lines 326-328: "The CLAAS-V002E1 data is computed using the MSG SEVIRI satellite channels as well as derived products and additional data. Each of them bring their own bias, potentially multiplying their effects on the final CTH. In contrast, our CTH is based only on the predicted reflectivity. In that way, we can minimize the influence of additional data sources."

**3.37 Comment (Figure 6)**

- **Comment**: Is the comparison calculated across the entire FD of the model predictions, or just the tracks that overlap with CloudSat? If it's non-normalized frequencies, they should be calculated across the same number of datapoints to be comparable, if I am not mistaken? Please clarify in the caption and main text.

- **Author's response**: Thank you for that remark, we will check the figure concerning the number of datapoints and revise the corresponding text.

- **Author's changes in the manuscript**: Figure 6 is now Figure 8 (page 20). The datapoints are described in Section 3.4, lines 286ff. and in the caption of Figure 8: "To compare the quality of the Res-UNet predictions, we calculate the CTH. We derive the CTH on the test dataset using the CloudSat reflectivities and the Res-UNet predictions.".

**3.38   Comment (Discussion)**

- **Comment**: I found it at times hard to follow when you are referring to your own work, versus work done by other people. Could you please go through this section again?

- **Author's response**: We revise the manuscript with particular attention to this issue.

- **Author's changes in the manuscript**: We revised Section 4 to be more clear and concise.

**3.39   Comment (line 280)**

- **Comment**: Please clarify what you mean by "minimal architecture".

- **Author's response**: This study is based on a small and rather standard architecture (UNet added by residual connections). We want to point out the possibility to interpolate continuous 3D clouds from the 2D data sources. That is why we stick to the UNet instead of using a more complex and powerful architecture (like ViT). We rewrite the sentence for clarity.

- **Author's changes in the manuscript**: Section 4., lines 337–340: "The Res-UNet shows a 30 % improvement in the mean RMSE (Fig. 4). We could potentially further enhance model performance by utilizing a more complex architecture. Our input data differs from typical gray-scale or RGB images, as it comprises multiple input channels and results in a 3D output (Drönner et al., 2018). Given the demands of our data and resource constraints, we adapted a standard UNet architecture rather than using a pre-trained model (Amato et al., 2020).".

**3.40   Comment (line 282)**

- **Comment**: "others" ==> others' work

- **Author's response**: Changed in the revised version.

- **Author's changes in the manuscript**: Section 4, lines 346–347: "In contrast to studies of Leinonen et al. (2019) or Hilburn et al. (2020), we incorporate a diverse landscape into our training.".

**3.41   Comment (line 284)**

- **Comment**: "Nonetheless, defining those variables as additional predictors has a negligible effect on the model performance." Are you showing evidence for this somewhere?

- **Author's response**: Similar explanation as for lines 264+. We tried a different model setup. Since it achieved poor results, we did not add it in the initial submission for brevity. In the revised manuscript, we change that paragraph.

- **Author's changes in the manuscript**: We changed the text in Section 4, lines 360–363: "In this study, we only derive daytime predictions. This is due to the influence of solar radiation in visible spectrum (VIS) channels (Hilburn et al., 2020; Jeppesen et al., 2019). Additional distortions may arise from VIS channels, as satellite images only represent the uppermost cloud layer. Depending on the location, they can be highly influenced by the surface albedo (Drönner et al., 2018).".

 ### 3.42    Comment (line 288)

- **Comment**: "Leaving out the affected channels downgrades the overall performance." Same here, do you have evidence supporting this statement?

- **Author's response**: Same explanation as above, will also be changed.

- **Author's changes in the manuscript**: See Comment 3.41. We changed the paragraph in Section 4 and added
 citations for the assumed effect of VIS channels.

**4    Referee 3**

**4.1    Comment (general)**

- **Comment**: This looks to me as if this study has the characteristics of an ML-based retrieval, where to properties being retrieved from the SEVIRI radiances are the vertical profile of radar reflectivity. This has some similarities
 to other algorithms, such as the GPROF retrieval (Kummerow et al, J. App.  Met., 2001), which could be discussed. Perhaps the method in this paper could produce improved precipitation retrievals in a similar style to the GPROF retrieval (although not in this publication)?

- **Author's response**: That is an interesting suggestion worth to be investigated. We see the similarity to reflectivity-based methods for precipitation retrieval and thought about this aspect before. A detailed analysis seemed out
 of scope for this paper. Nevertheless, we think this would be an interesting follow-up.

- **Author's changes in the manuscript**: We did not change the manuscript based on this comment. We rather think this could be an interesting suggestion for our future work.

**4.2    Comment (general)**

- **Comment**: Although it doesn't seem to be at the level where I could be comfortable using the retrieved
 reflectivity profiles yet, but perhaps there are certain conditions where the data could be more trustworthy? Would limiting the study to a smaller area (e.g. close to nadir for SEVIRI) produce better results?

- **Author's response**: We agree with you on this issue. In our paper, we state the current problems with the method and its limits in terms of accuracy (Section 4). Resampling the satellite data on a spatial grid with geographic coordinates affects the accuracy of the input data with greater distance to the equator (Section 2.1.1). We think
 limiting the area to e.g. the tropics has the potential to improve the results. In the revised manuscript, we add a visualization of the zonal error to display the geographic differences.

- **Author's changes in the manuscript**: We added a zonal analysis of the RMSE and CTH (Figure 7 (a), 10 (b)) to visualize the model performance in different regions of the FD. The results are described in Section 3.3 and 3.4. We add a statement in Section 4, lines 364–366: "Reducing the extent of the AOI can mitigate these geographic
 differences but limits the applicability of the network. Training regional models and adjusting the loss function and model architectures offer potential solutions to improve the results of the 3D cloud tomography.".

**4.3    Comment (line 9)**

- **Comment**: Bias in retrievals based on approximations and cloud physical properties L9 - It is later stated that RMSE varies significantly by cloud situation - how is this value derived?

 - **Author's response**: We did not classify the cloud type, but we investigated the RMSE on the test set with 1500 samples. We rephrase that sentence for clarity.

- **Author's changes in the manuscript**: We describe the results of the 2D reconstruction in Section 3.2. We change the abstract, lines 4–5: "The Res-UNet extrapolates 2D reflectivities across the satellite's full disk, enabling a reconstruction of the cloud intensity, height, and shape in 3D.".

**4.4 Comment (line 10)**

- **Comment**: receive high accordance -> find high agreement?

- **Author's response**: Thank your that remark, will be changed.

- **Author's changes in the manuscript**: Abstract, lines 7–8: " While the model aligns well with CloudSat data, it simplifies multi-level and mesoscale clouds, in particular.".

**4.5 Comment (line 30)**

- **Comment**: at risk of bias. I would argue this is almost always an issue (and would be for a ML method too)

- **Author's response**: You are right, no method is probably completely free of bias. We will rephrase that sentence.

- **Author's changes in the manuscript**: We omit that statement in Section 1, lines 19–21: "By using the satellite's specificity at different wavelengths (Thies and Bendix, 2011), and subjective labeling or fixed thresholds (Platnick et al., 2017), we can estimate cloud physical properties like the cloud optical thickness (Henken et al., 2011) or the effective radius (Chen et al., 2020).".

**4.6 Comment (line 30)**

- **Comment**: If passive sensors lack this information, how can it be recovered from this method?

- **Author's response**: Information on the cloud column below the cloud top can be approximated using the satellite channels at different wavelengths (lines 27-29). While they enable an approximation, their information density is lower compared to the radar reflectivity received from active radar. The hypothesis is that the information on the vertical resolution is hidden within the satellite channels. While previous research reached their limits, we assume that these representations can be learned from a neural network better than by human analysis. We rephrase that paragraph for clarity.

- **Author's changes in the manuscript**: Section 1, lines 25–26: "Passive sensors can be used to deliver an approximation of the cloud vertical column, but their information density is reduced compared to active sensors (Noh et al., 2022).".

**4.7 Comment (line 33)**

- **Comment**: joined -> combined

- **Author's response**: Changed to: "A combined use of different instruments to derive comprehensive 3D structures [...]".

- **Author's changes in the manuscript**: Section 1, lines 26–27: "Combining data sources can fill current data gaps (Amato et al., 2020; Steiner et al., 1995). The combined use of different instruments has been investigated before.".

**4.8 Comment (line 88)**

- **Comment**: The near IR channels also include reflected solar radiation (depending on channel wavelength and time of day)

- **Author's response**: We rephrase that sentence to clarify the channel differences.

- **Author's changes in the manuscript**: Section 2.1.1, lines 78–79: "Depending on the wavelength and daytime of retrieval, the channels are sensitive to reflected solar radiation or surface emissions.", and Table 1.

**4.9 Comment (line 109)**

- **Comment**: How is parallax addressed? SEVIRI has an off-nadir view for most points on the disc, while CloudSat always views at nadir. This could cause registration issues between different cloud layers. Changes in resolution might also become an issue closer to the edge of the disc.

- **Author's response**: We recognize the importance of the parallax effect for the correction of the geometry, especially when dealing with multiple cloud layers. In this work, we estimate only a small impact on the overall results. First, we assume that neighbouring pixels are more similar to each other than those in far distance. Second, we face the need to downsample the resolution of the CloudSat data to match the MSG SEVIRI resolution. That is why we don't expect the estimated differences of up to 3 pixels to substantially impact the predicted reflectivities. Overall, we estimate the error resulting from geometric inaccuracy to be lower than current model limitations. We will add a remark in the manuscript.

- **Author's changes in the manuscript**: We add a statement in Section 4, lines 353–356: "We emphasize the influence of the geographic location and the CloudSat orbit, particularly in regions farther from the equator, where sensor accuracy diminishes (Fig. 7). Currently, we estimate the model error to be mainly influenced by the data imbalance and chosen loss function. In the future, addressing the cloud parallax shift in high-angle satellite observations could enhance the results by a more accurate image-profile matching (Bielinski, 2020.)".

**4.10 Comment (line 110)**

- **Comment**: I found the last sentence difficult to follow. What does the 'factor to coarse grain the data' mean here?

- **Author's response**: The resolution mismatch between CloudSat and MSG SEVIRI requires an aggregation of CloudSat pixels. We use the maximum reflectivity to reduce the resolution of the radar data. This blurs the sharp contrasts within the radar cross sections (as seen in Figure 4, CloudSat CPR cross sections are blurred compared to the original data). We rephrase that sentence for clarity.

- **Author's changes in the manuscript**: Section 2.1.3, lines 106–109: "CloudSat flies across a horizontal transect within the satellite scene. It has a higher native resolution than MSG SEVIRI. To align the datasets, we downsample the radar pixels by aggregating them based on the local maximum reflectivity. This adjusts the CloudSat pixels to the MSG SEVIRI resolution of 0.03 ° but leads to some loss of sharp contrast in radar pixels (Jordahl et al., 2020).".

**4.11 Comment (Figure 2 )**

- **Comment**: Why not low cloud layers? This is explained later on, but could have been included in the description of the radar data (unless I missed it?)

- **Author's response**: As stated in lines 123-125, we analyzed the CloudSat quality flag to identify height levels that were affected by noise. This applies especially to the near-ground height levels influenced by the topography and connected radar attenuation. Those height levels were excluded from the model framework. The reduction affects the representation of low clouds and the cloud base, reducing the model performance in these altitudes. We add an extended explanation in this section in the revised manuscript.

- **Author's changes in the manuscript**: Figure 2 is now Figure 4. In Section 2.1.4, lines 122–125, we describe the modification of the CloudSat data: "As described in Section 2.1.2, we reduce the height levels of the CloudSat profile from 125 to 90 (Fig. 1, 2). We use the CloudSat quality index to identify noisy pixels. Pixels with a quality index lower than six were set to a background value of -25 dBZ to reduce noise (Marchand et al., 2008).". This includes the reduction of the height levels and filtering all values with a quality flag < 6. The analysis of the height levels can be found in Section 3.2, 3.4, and 4.

**4.12 Comment (Figure 3)**

- **Comment**: How is this normalisation done? Is it across the whole image?

- **Author's response**: For the joint plot, we computed the value distribution for observed and predicted reflectivities on the test set. The normalisation is done across the whole image using the respective distributions.

- **Author's changes in the manuscript**: Section 3.1, lines 217–219: "We analyze the difference between the observed and predicted reflectivities by a two-dimensional joint distribution plot. For this purpose, we calculate the density distribution of the reflectivity between 2.4–24 km. Here, we use a bin size of 1 dbZ and 240 m height, respectively (Steiner et al., 1995). All distributions are calculated on the test dataset and normalized by the distribution size (n = 1500)."

**4.13 Comment (line 220)**

- **Comment**: Why is the DL model different here? Is this due to a regression-to-the-mean effect for the Res-UNet?

- **Author's response**: After revising, we think this was due to an error in the code. The results could differ as a result of the regression-to-the-mean. Since the DL network is fed a lot of nearly cloudfree images, they can lead to a shift of the distribution. In contrast, extreme values have a higher influence on the OLS and RF. This can affect the prediction of positive values. Nevertheless, we change the plot to all models pointing towards the same direction.

- **Author's changes in the manuscript**: In the revised version of Figure 3, the directions are the same for the Res-UNet, the OLS, and the RF (page 12).

**4.14 Comment (line 262)**

- **Comment**: I think thin clouds are also difficult to detect from CloudSat, given they might have a small radar reflectivity. Both SEVIRI and CloudSat on their own are able to produce this second, higher peak in cloud top height, suggesting they can detect these thinner clouds.

- **Author's response**: For the first sentence, we agree and rephrase that statement. The second part I found to be contradicting. CloudSat in particular can produce this peak whereas our model fails to appropriately reconstruct high reflectivities in higher altitudes (Fig.3). The missing peak might be connected to the imbalance within the data and resulting underestimation of high reflectivities. We rephrase that paragraph for clarity.

- **Author's changes in the manuscript**: In Section 3.1, we describe the reflectivity distribution per height level. Lines 223–225: " Since we observe few clouds in high altitudes (Fig. 2, a), the distribution differences get smaller above 15 km.". In Section 3.2, lines 235–238: "Above 15 km, we have few CloudSat observations > 15 dBZ (Fig. 2, a). The lack imbalance of the CloudSat data can affect the predicted CTH. This is described in Section 3.4, lines 294–299: "The shift of the distribution is reflected in the CTH in Figure 8 (b). Both datasets display a maximum CTH at 7 km height. This first peak is overestimated by the model. A second peak around 12–15 km height is underestimated by the Res-UNet. The difference between the predicted CTH is reflected within Figure 8 (d). The mismatch between the two peaks is about the same size. The underestimated second peak can be led back to the inaccuracy of the predicted reflectivities. The Res-UNet overestimates reflectivities < -15 dBZ at all height levels up to 15 km. It misses high reflectivities responsible for the peak of the CTH at 12-–15 km (Fig. 3, b). Instead, we see an overall surplus of background values in the FD prediction.".

**4.15 Comment (Figure 4)**

- **Comment**: Perhaps blur the Cloudsat data to match the DL resolution? It would also be nice to have 5km marked at the bottom of the y-axis (if that is the case), to highlight this doesn't go to zero.

- **Author's response**: Figure 4 shows the blurred CloudSat data after downsampling. The models predict the cross section with a higher blurriness, as a result of the regression-to-the-mean. We will add the lower y-axis label.

- **Author's changes in the manuscript**: Figure 4 is now Figure 5 (page 15). Lower y-axis labels are added.

**4.16 Comment (line 270)**

- **Comment**: So it is (or could be) a simultaneous retrieval, due to the inclusion of the water vapour channels?

- **Author's response**: We agree the water vapour channels could lead to this distortion over water bodies. There are only few observations in these regions. Extreme values have a higher influence on the mean. This could be a further source for distortion.

- **Author's changes in the manuscript**: A geographical analysis of the CTH can be found in Section 3.4, lines 313–316: "This issue is reflected within Figure 10. Here, we visualize the geographic distribution of the CTH difference (Fig. 10, a). The mean difference over all pixels accounts for 1.28 km. While the data show an overall agreement, the pixel-wise difference rises to a maximum of 10 km. This applies especially to regions in the subtropics. We observe an underestimation of the predicted CTH over land. Above the Atlantic ocean, especially in the tropics, the predictions are too high. The highest difference occurs in subtropics on the northern hemisphere (Fig. 10, b).", and Section 4, lines 356–358: "The most accurate predictions fall between 25 °N and 25 °S (Fig. 7), while mid-and high-latitudes exhibit a higher RMSE. This is likely due to the land-sea distribution and connected cloud patterns. Over ocean bodies, the model overestimates the reflectivity (Fig. 10, a). Using the water vapor channels could lead to this distortion.".

**4.17 Comment (Figure 7)**

- **Comment**: Much more structure on the Res-UNet results - why is this? Also, the Res-UNet image seems to have almost all the cloud tops at the same altitude, other than a small fraction of low-level cloud in some regions.

- **Author's response**: Both are aggregated to a monthly mean, whereas the datasets have a resolution mismatch (0.05 ° for CLAAS-V002E, 0.03 ° for the predictions). The aggregation of the derived CTH contains a higher small-scale variability. The results reflect the lacking model ability to for predict low level and high level clouds (Fig.3). Using a fixed threshold of -15dBZ to identify clouds may not be accurate for all regions of the FD. Combining the threshold with the shifted reflectivity distribution can affect the distribution of the CTH in, smearing out regional differences. As a result of this underestimation of high reflectivities, the mean of the monthly aggregate is shifted towards a lower CTH than for the CMSAF product. CloudSat information is retrieved at the same local time for every region (due to its sun-synchronous orbit) and only applicable to daytime predictions. This can increase the diurnal representation of the reflectivities.

- **Author's changes in the manuscript**: The description of the results for the CTH can be found in Section 3.4, lines 303–313: "For each time step, we calculate the CTH on the FD and aggregate the results to a monthly mean. These values are compared to the CLAAS-V002E1 product with a resolution of 0.05 ° (Finkensieper et al., 2020). The predicted CTH has a resolution of 0.03 °. Due to this mismatch, our predictions show more fragmented structures (Sect. 2.3.3). Although the small-scale accuracy is improvable, derived data and deducted parameters allow an investigation of regional differences. The analysis reveals an overall high agreement. Regional differences arise around the equator and mid- to high-latitudes (Fig. 9). In mid-latitudes, the CTH over water bodies is overestimated in the southern hemisphere and underestimated in the northern hemisphere. These differences can be led back to an increased RMSE in these regions (Fig. 7). In contrast, a low RMSE in the subtropics increases the accuracy of the predicted CTH. The model is biased toward predicting lower clouds than the observational data. Overall, the Res-UNet overestimates the occurrence of clouds in 6-8 km while underestimating high clouds (Fig. 8, b).".

**4.18 Comment (line 299)**

- **Comment**: I guess this is much like satellite retrievals in general? Presumably there is actually just a lack of information there?

- **Author's response**: Apart from the inaccuracy of the model results itself, we agree this could be traced back to a general lack of information. We rephrase the sentence to clarify.

- **Author's changes in the manuscript**: Section 4, lines 369ff.: "A common limitation is accurately representing multi-layer clouds. Using the satellite channels to derive these information may be limited (Schmetz et al., 2002;

Thies and Bendix, 2011). High reflectivities tend to be underestimated due to noise near the ground (Stephens et al., 2008). To mitigate this, we exclude affected height levels, but this results in incomplete model predictions between 0—5 km (Fig. 2).".

**4.19   Comment (line 300)**

- **Comment**: 5km is quite high from the ground to deal with clutter. It also cuts out a lot of the lower level clouds with a more challenging cloud top height retrieval, which might make the DL method seem better overall? I am not suggesting they should be included (working with just high clouds is an important task), but could be worth mentioning.

- **Author's response**: We are aware that low level clouds are important features needed to be represented within the model results. CloudSat suffers from attenuation in low height levels. After analyzing the CloudSat quality flag, we observe that most of the reflectivity values between 0–2.4 km are affected by noise, so we set these values to -25 dBZ. This affects the predictions between 2.4 and 5km height. Due to the regression-to-the-mean and the imbalanced reflectivity distribution, we see a further shift of the distribution towards low reflectivities, especially up to 5 km. Using a modified network architecture and loss function can help to improve the representation of clouds between 2.4–5km, and the calculation of the CTH. We include a statement in the revised manuscript.

- **Author's changes in the manuscript**: This issue is described in Section 2.1.2 and Section 4, lines 367–374: "The reconstructed CloudSat cross sections are comparable to results achieved by Leinonen et al. (2019). For both studies, the RMSE varies between 0–1 dBZ for cloud-free samples, 3–7 dBZ for more uniform clouds, and more than 10 dBZ for multi-layer clouds. A common limitation is accurately representing multi-layer clouds. Using the satellite channels to derive these information may be limited (Schmetz et al., 2002; Thies and Bendix, 2011). High reflectivities tend to be underestimated due to noise near the ground (Stephens et al., 2008). To mitigate this, we exclude affected height levels, but this results in incomplete model predictions between 0—5 km (Fig. 2). Reducing noise is crucial for improving the performance of DL applications in remote sensing (Enitan and Ilesanmi, 2021). Our results are significantly influenced by the resolution difference between CloudSat and MSG SEVIRI, as well as the choice of the loss function (Sect. 2.1.3).".

**4.20   Comment (line 321)**

- **Comment**: I think this reduction in bias should be shown if it is claimed. Those external data sources might decrease bias themselves in some situations (e.g. with a better estimate of water vapour than is available from the SEVIRI IR channels).

- **Author's response**: The statement is drawn from the comparison of the CMSAF CTH and the predicted CTH, but we revise the paragraph and add a further quantification.

- **Author's changes in the manuscript**: Section 5, lines 396–399: "Using only the predicted reflectivity, we derive the CTH without external data sources. Overall, the approach accurately reconstructs cloud structures in varying environmental conditions on the FD. Although the results are affected by sensor-specific and technical limitations, a vast potential for applications in atmospheric and climate sciences is apparent.".

**4.21   Comment (line 325)**

- **Comment**: maybe "remote oceanic regions" instead of "secluded regions above the sea surface"?

- **Author's response**: We appreciate that suggestion and change the wording here.

- **Author's changes in the manuscript**: Section 5, lines 400–401: "We emphasize the benefit to extrapolate a 3D cloud field, especially in remote oceanic regions.".

---

## Author Response (AR2)

**Author's response - minor revisions**
**(Manuscript "egusphere-2023-1834")**

Sarah Brüning, Stefan Niebler, and Holger Tost

December 13, 2023

**1  Report 1 / Referee 3**

**1.1  Comment (L 76)**

- **Comment**: L76 - The CloudSat data is also 'satellite data' how about visible/infrared, VIS/IR or imager data?

- **Author's response**: We change the term "satellite data" to "imager data" throughout the text.

- **Author's changes in the manuscript**: Lines 76–77: "Satellite images from the MSG SEVIRI instrument displays the input for the network (later referred to as "imager data") (Schmetz et al., 2002).".

**1.2  Comment (Table 1)**

- **Comment**: Table 1 - Aren't all the solar channels daytime only? I see what you are trying to say, but perhaps the IR3.9 channel should be 'partial'? Alternatively, another name for the column might be more appropriate, perhaps 'Type' which could then be 'Solar reflective', 'Thermal IR' or 'Both'?

- **Author's response**: We see your point and change the naming of the column.

- **Author's changes in the manuscript**: Page 4, Table 1, last column: "Type".

**1.3  Comment (L 101)**

- **Comment**: L101 - perhaps 'the AOI of SEVIRI' (rather than 'the satellite'), or even just 'the AOI'?

- **Author's response**: We see this is misleading, changed to "the AOI".

- **Author's changes in the manuscript**: Line 100–102: "To match the datasets, we compare their timestamps and locations. If the radar coordinates fall within the AOI, we determine the flight direction to identify whether CloudSat circles the Earth in ascending or descending orbit.".

**1.4  Comment (L 304)**

- **Comment**: L304 - How closely would you expect these datasets to match? I don't think you need to add much here, but given the CloudSat radar often misses thin clouds, should there be a close match between these datasets?

- **Author's response**: We expect an overall good match whereas our results are less sensitive to thin clouds. The similarity is expected to decrease especially in low and high altitudes where the CloudSat radar is affected by sensor limitations. This height-dependent variability can influence the overall results which reduces the dataset agreement.

- **Author's changes in the manuscript**: Lines 307–309: "CloudSat faces sensor limitations in low and high altitudes of the troposphere (Sect. 2.1.2). While our analysis reveals an overall high agreement, the lack of e.g. thin clouds within the radar data affects the similarity between the CLAAS-V002E1 data and the predicted CTH."

**1.5 Comment (L 308)**

- **Comment**: L308 - This wording suggests that it is the hemisphere that is important. What would be the reason for this? Is it a seasonal cycle in cloud types, the solar zenith angle, or perhaps something else?

- **Author's response**: Although we have few observations in the tropics on the southern hemisphere, the model performs better than in the northern hemisphere. The distribution of land masses affects the formation of clouds in both hemispheres. This leads to a varying model performance depending on the cloud type. The seasonal cycle and solar zenith angle can further increase the difference.

- **Author's changes in the manuscript**: Lines 310–316: "We observe a connection between the similarity of the datasets and the hemisphere. In the northern hemisphere, the highest amount of image-profile pairs and the highest CTH difference occur between 0–20 °N. Between the tropics of the southern hemisphere, the amount of observations is similar whereas the CTH difference is considerably lower. The variability between the hemispheres can be led back to the distribution of land masses. A higher proportion of oceans in the southern hemisphere and the solar zenith angle affect the formation of clouds (Bruno et al., 2021). The result is an increased model performance which might be caused by the existence of either more uniform or less complex clouds.".

**1.6 Comment (L 330)**

- **Comment**: L330 - As above, I would avoid using the term 'satellite' here.

- **Author's response**: We change the term to "imager data".

- **Author's changes in the manuscript**: Lines 334–335: "The CLAAS-V002E1 data is computed using the MSG SEVIRI imager data as well as derived products and additional data.".

**1.7 Comment (L 391)**

- **Comment**: L391 - Such as aerosol measurements - this would depend on the information content of the imager data. It is not clear to me that there is information in these measurements to do a full 3D retrieval of aerosol, even with deep learning techniques (although I would love to be proved wrong!)

- **Author's response**: We do not know for sure whether the approach can be used to extrapolate 2D aerosol measurements to a 3D perspective. Former studies derived e.g. the aerosol optical thickness and type from satellite data (e.g. https://doi.org/10.1029/2009JD012272). We think evaluating the possibility of a full 3D retrieval of aerosol received by e.g. aircraft measurements would be an interesting follow-up.

- **Author's changes in the manuscript**: Lines 398–401: " While we use CloudSat radar data as our ground truth, it needs to be evaluated whether this approach can be adapted to other 2D transect data sources, such as aerosol measurements. Former studies already derived aerosol properties from imager data (Carrer et al., 2010). The DL framework could help to achieve a full 3D retrieval of aerosols.".

**2 Report 2 / Referee 2**

**2.1 Comment (L 376)**

- **Comment**: Line 376: The last citation in the following sentence seems misplaced: "In contrast, Leinonen et al. (2019) use data from the MODIS satellite (Zantedeschi et al., 2022)."

- **Author's response**: Thank you for pointing this out, we remove the misplaced citation.

- **Author's changes in the manuscript**: Lines 384–385: "In contrast, Leinonen et al. (2019) use data from the MODIS satellite.".